



# Soil related developments of the Biome-BGCMuSo v6.2 terrestrial ecosystem model by integrating crop model components

Dóra Hidy[1,2*], Zoltán Barcza[1,3,4], Roland Hollós[1,4], Laura Dobor[5], Tamás Ács[6], Dóra Zacháry[7], Tibor Filep[7], László Pásztor[8], Dóra Incze[3], Katarína Merganičová[4,9], Peter Thornton[10], Steven Running[11], Nándor Fodor[4]

[1] Excellence Center, Faculty of Science, ELTE Eötvös Loránd University, H-2462 Martonvásár, Hungary

[2] Agroecology Recearch Group, MTA-MATE, H-2100 Gödöllő, Hungary

[3] Department of Meteorology, Institute of Geography and Earth Sciences, ELTE Eötvös Loránd University, H-1117 Budapest, Hungary

[4] Faculty of Forestry and Wood Sciences, Czech University of Life Sciences Prague, 165 21 Prague, Czech Republic

[5] Centre for Agricultural Research, Agricultural Institute, H-2462 Martonvásár, Hungary

[6] Department of Sanitary and Environmental Engineering, Budapest University of Technology and Economics, H-1111 Budapest, Hungary

[7] Geographical Institute, Researche Centre for Astronomy and Earth Sciences, H-1112 Budapest, Hungary

[8] Institute for Soil Sciences, Centre for Agricultural Research, H-1022 Budapest, Hungary

[9] Department of Biodiversity of Ecosystems and Landscape, Institute of Landscape Ecology, Slovak Academy of Sciences, SK 949 01 Nitra, Slovakia

[10] Climate Change Science Institute/Environmental Sciences Division, Oak Ridge National Laboratory, Oak Ridge, TN 37831, USA

[11] Numerical Terradynamic Simulation Group, Department of Ecosystem and Conservation Sciences University of Montana, Missoula, MT 59812, USA

*Correspondence to*: Dora Hidy (dori.hidy@gmail.com)



## Abstract

Terrestrial biogeochemical models are essential tools to quantify climate-carbon cycle feedback and plant-soil relations from local to global scale. In this study, theoretical basis is provided for the latest version of Biome-BGCMuSo biogeochemical model (version 6.2). Biome-BGCMuSo is a branch of the original Biome-BGC model with a large number of developments and structural changes. Earlier model versions performed poorly in terms of soil water content (SWC) dynamics in different environments. Moreover, lack of detailed nitrogen cycle representation was a major limitation of the model. Since problems associated with these internal drivers might influence the final results and parameter estimation, additional structural improvements were necessary. During the developments we took advantage of experiences from the crop modeller community where internal process representation has a long history. In this paper the improved soil hydrology and soil carbon/nitrogen cycle calculation methods are described in detail. Capabilities of the Biome-BGCMuSo v6.2 model are demonstrated via case studies focusing on soil hydrology and soil organic carbon content estimation. Soil hydrology related results are compared to observation data from an experimental lysimeter station. The results indicate improved performance for Biome-BGCMuSo v6.2 compared to v4.0 (explained variance increased from 0.121 to 0.8 for SWC, and from 0.084 to 0.46 for soil evaporation; bias changed from -0.047 to -0.007 $m^3$ $m^{-3}$ for SWC, and from -0.68 mm $day^{-1}$ to -0.2 mm $day^{-1}$ for soil evaporation). Sensitivity analysis and optimization of the decomposition scheme is presented to support practical application of the model. The improved version of Biome-BGCMuSo has the ability to provide more realistic soil hydrology representation and nitrification/denitrification process estimation which represents a major milestone.



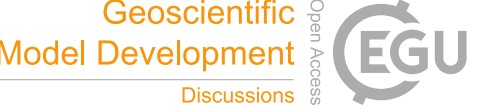

## 1. Introduction

The construction and development of biogeochemical models (BGM) is the response of the scientific community to address challenges related to climate change and human induced global environmental change. BGMs can be used to quantify future climate-vegetation interaction including climate-carbon cycle feedback, and as they simulate plant production, they can be used to study a variety of ecosystem services that are related to human nutrition and resource availability (Asseng et al., 2013; Bassu et al., 2014; Huntzinger et al., 2013). Similarly to the models describing various and complex environmental proscesses, the structure of biogeochemical models reflects our current knowledge about a complex system with many internal processes and interactions.

Processes of the atmosphere-plant-soil system take place on different temporal (sub-daily to centennial) scales and are driven by markedly different mechanisms that are quantified by a large diversity of modeling tools (Schwalm et al., 2019). Plant photosynthesis is an enzyme-driven biochemical process that has its own mathematical equation set and related parameters (and a large literature; e.g. Farquhar et al., 1980; Medlyn et al. 2002; Smith and Dukes, 2013; Dietze, 2013). Allocation of carbohydrates in the different plant compartments is studied extensively and also has a large literature and mathematical tool set (Friedlingstein et al., 1999; Olin et al., 2015; Merganičová et al., 2019). Plant phenology is quantified by specific algorithms that are rather uncertain components of the models (Richardson et al., 2013; Hufkens et al., 2018; Peaucelle et al., 2019). Soil biogeochemistry is driven by microbial and fungal activity and also has its own methodology and a vast literature (Zimmermann et al 2007; Kuzyakov, 2011; Koven et al., 2013; Berardi et al., 2020). Emerging scientific areas like the quantification of the dynamics of non-structural carbohydrates (NSC) in plants has a separate methodology that claims for mathematical representation in models (Martínez-Vilalta et al., 2016). Simulation of land surface hydrology including evapotranspiration is typically handled by some variant of the Penman-Monteith equation that is widely studied thus represents a separate scientific field (McMahon et al., 2013; Doležal et al., 2018).

Putting all together, if we are about to construct and further improve a biogeochemical model to consider novel findings and track global changes, we need a comprehensive knowledge that integrates many, almost disjunct scientific fields. Clearly, transparent and well-documented development of a biogeochemical model is of high priority but challenging


from the very beginning that claims for cooperation of researchers from various scientific
fields.
Continuous model development is inevitable but it has to be supported by extensive
comparison with observations and some kind of implementation of the model-data fusion
approach (Keenan et al., 2011). It is well documented that structural problems might trigger
incorrect parameter estimation that might be associated with distorted internal processes
(Sándor et al., 2017; Martre et al., 2015). In other words, one major issue with BGMs (and in
fact with all models using many parameters) is the possibility to get good simulation results
for wrong reasons (which means incorrect parameterization) due to compensation of errors
(Martre et al., 2015). In order to avoid this issue, any model developer team has to make an
effort to focus also internal ecosystem conditions (e.g. soil volumetric water content (SWC),
nutrient availability, stresses, etc.) and other processes (e.g. decomposition) rather than the
main simulated processes (e.g. photosynthesis, evapotranspiration).
Historically, biogeochemical models have been developed to simulate the processes of
undisturbed ecosystems with simple representation of the vegetation (Levis, 2010). As the
focus was on the carbon cycle, water and nitrogen cycles and related soil processes were not
well represented. Incorrect representation of SWC dynamics is still an issue with the models
especially in drought-prone ecosystems (Sándor et al., 2017). Additionally, human
intervention representation (management) was missing in many cases and it seems that some
state-of-the-art BGMs still lack the representation of e.g. thinning, grass mowing, grazing,
fertilization, planting or harvest (Table A1 in Friedlingstein et al., 2010).
In contrast, crop models with different complexity were used for about 50 years or so
to simulate the processes of managed vegetation (Jones et al., 2017; Franke et al., 2020). As
the focus of the crop models is on final yield due to economic reasons, the carbon balance, or
the full greenhouse gas balance was not, or was just partially addressed originally. Crop
models typically have a sophisticated representation of soil water balance with a multilayer
soil module that usually calculates plant response to water stress as well. Nutrient stress, soil
conditions during planting, consideration of multiple phenological phases, heat stress during
anthesis, vernalization, manure application, fertilization, harvest, and many other processes
have been implemented during the decades (Ewert et al., 2015). Therefore, it seems to be
straightforward to exploit the benefits of crop models and implement sound and well-tested
algorithms into the BGMs.
Our group has been developing Biome-BGCMuSo for 15 years. Starting from the
well-known Biome-BGC model originally developed to simulate undisturbed forests and





grasslands, using a simple single layer soil submodel (Running and Hunt, 1993; Thornton and
Rosenbloom, 2005), we developed a complex, more sophisticated model (Hidy et al., 2012;
2016). Biome-BGCMuSo v4.0 (Biome-BGC with Multilayer Soil module) uses a 7-layer soil
module and is capable of simulating different ecosystems from natural grassland to cropland
including several management options, taking into account many environmental effects (Hidy
et al., 2016). The developments included improvements regarding both soil and plant
processes. In a nutshell, the most important, soil related developments were the improvement
of the soil water balance module by implementing routines for estimating percolation,
diffusion, pond water formation and runoff; the introduction of multilayer simulation  for
belowground processes in a simplified way. The most important, plant related developments
involved the implementation of a routine for estimating the effect of drought on vegetation
growth and senescence; the improvement of stomatal conductance calculation considering
atmospheric $CO_2$ concentration; the integration of selected management modules; the
implementation of new plant compartments (e.g. yield); the implementation  of C4
photosynthesis routine; the implementation of photosynthesis and respiration acclimation of
plants and temperature-dependent Q10; and empirical estimation of methane and nitrous
oxide soil efflux.

Problems found with the Biome-BGCMuSo v4.0 simulation result (such as the poor

representation of soil water content (Hidy et al., 2016; Sándor et al., 2017) or the lack of
sophisticated, layer-specific soil nitrogen dynamics representation) and the model structure
related problems (such as the lubber parameterization of the model) marked the path for
further developments.

The aim of the present study is to provide detailed documentation on the current,

improved version of Biome-BGCMuSo v6.2, which has many new features and facilitates
various in-depth investigations of ecosystem functioning. Due to large number of
developments, this paper focuses only on the soil related model improvements. Case studies
are also presented to demonstrate the capabilities of the new model version and to provide
guidance for the model user community.

## 2. The original Biome-BGC model

Biome-BGC was developed from the Forest-BGC mechanistic model family in order

to simulate vegetation types other than forests. Biome-BGC was one of the earliest



biogeochemical models that included explicit carbon and nitrogen cycle modules. Biome-
BGC simulates the storages and fluxes of water, carbon, and nitrogen within and between the
vegetation, litter, and soil components of terrestrial ecosystems. It uses a daily time step, and
is driven by daily values of maximum and minimum temperatures, precipitation, solar
radiation, and vapor pressure deficit (Running and Hunt, 1993). The model calculations apply
to a unit ground area that is considered to be homogeneous.
The three most important components of the model are the phenological, the carbon
uptake and release, and the soil flux modules. The core logic that is described below in this
section remained intact during the developments. The phenological module calculates foliage
development that affects the accumulation of C and N in leaf, stem (if present), root and
consequently the amount of litter. In the carbon flux module gross primary production (GPP)
of the biome is calculated using Farquhar's photosynthesis routine (Farquhar et al., 1980) and
the enzyme kinetics model based on Woodrow and Berry (2003). Autotrophic respiration is
separated into maintenance and growth respiration. Maintenance respiration is calculated as
the function of the N content of living plant pools, while growth respiration is an adjustable
but fixed proportion of the daily GPP. The single-layer soil module simulates the
decomposition of dead plant material (litter) and soil organic matter, N mineralization and N
balance in general (Running and Gower, 1991). The soil module uses the so-called
converging cascade method (Thornton and Rosenbloom, 2005) to simulate decomposition,
carbon and nitrogen turnover, and related soil $CO_2$ efflux.
The simulation has two basic steps. During the first (optional) spinup simulation the
available climate data series is repeated as many times as it is required to reach a dynamic
equilibrium in the soil organic matter content to estimate the initial values of the carbon and
nitrogen pools. The second, normal simulation uses the results of the spinup simulation as
initial conditions and runs for a given, predefined time period (Running and Gower 1991). So-
called transient simulation option (which is the extension of the spinup routine) is a novel
feature in Biome-BGCMuSo in order to ensure smooth transition between the spinup and
normal phase (Hidy et al., 2021).
In Biome-BGC, the main parts of the simulated ecosystem are defined as plant, litter
and soil. The most important pools include leaf (C, N and intercepted water), root (C, N),
stem (C, N), soil (C, N and water) and litter (C, N). Plant C and N pools have sub-pools
(actual pools, storage pools and transfer pools). The actual sub-pools store C and N for the
current year growth. The storage sub-pools (essentially the non-structural carbohydrate pool,
the source for the cores or buds) contain the amount of C and N that will be active during the





next growing season. The transfer sub-pools inherit the entire content of the storage pools at
the end of every simulation year. Soil C also has sub-pools representing various organic
matter forms characterized by considerably different decomposition rates.

In spite of its popularity and proven applicability, the development of Biome-BGC
was temporarily stopped (the latest official NTSG version is Biome-BGC 4.2;
https://www.ntsg.umt.edu). One major drawback of the model was its relatively poor
performance in modelling managed ecosystems, and the simplistic soil water balance
submodel using a single soil layer only.

Our team started to develop the Biome-BGC model further in 2006. According to the
logic of the team, the new model branch was planned to be the continuation of the Biome-
BGC model with regard to the original concept of the developers (keeping the model code
open source, providing detailed documentation, and providing support for the users).

The starting point of our model development was Biome-BGC v4.1.1 that was a result
of the model improvement activities of the Max Planck Institute (Vetter et al., 2007).
Development of the Biome-BGCMuSo model branch has a long history by now. Previous
model developments were documented in Hidy et al. (2012) and Hidy et al. (2016). Below,
we provide detailed description of the new developments that are included in Biome-
BGCMuSo v6.2 which is the latest version released in September, 2021. A comprehensive
review of the input data requirement of the model together with explanation on the input data
structure is available in the User's Guide (Hidy et al., 2021). In this paper we refer to some
input files (e.g. soil file, plant file) that are described in the User's Guide in detail.

One of the most important novelty and advantage of the new model version (Biome-
BGCMuSo v6.2) compared to any previous versions that due to the extensive and detailed
soil parameter set (current version has 79, MuSo 4.0 has 39 and original model version has
only 6 adjustable soil related parameters) the parameterization of the model is much more
flexible. But this might be of course a challenging task to define all of the input parameters. In
order to support practical application of the model, the User's Guide contains proposed values
for most of the new parameters (Hidy et al., 2021).

## 213 3. Soil hydrology related developments

In Biome-BGCMuSo v6.2 a 10-layer soil submodel was implemented. Previous model
versions included a 7-layer submodel, which turned out to be insufficient to capture
hydrological events like drying of the topsoil layers with sufficient accuracy. The thicknesses





of the layers from the surface to the bottom are 3, 7, 20, 30, 30, 30, 30, 50, 200 and 600 cm.
The centre of the given layer represents the depth of each soil layer. Soil texture can be
defined by the percentage of sand and silt for each layer separately along with the most
important physical and chemical parameters (pH, bulk density, characteristic SWC values,
drainage coefficient, hydraulic conductivity) in the soil input file (Hidy et al., 2021).
The water balance module of Biome-BGCMuSo has five major components to
describe soil water related processes in daily resolution (listed here following the order of
calculation): pond water accumulation and runoff; infiltration and downward gravitational
flow (percolation); water potential gradient driven water movement within the soil (diffusion);
evaporation and transpiration (root water uptake); and the downward/upward fluxes to/from
groundwater. In the following subsections these five major components are described.

### 3.1 Pond water accumulation and runoff

Precipitation can reach the surface as rain or snow (below 0 °C snow accumulation is
assumed). Snow water melts from the snowpack as a function of temperature and radiation
and added to the precipitation input.
The canopy can intercept rain. The intercepted volume goes into the *canopy water*
pool, which can evaporate. No canopy interception of snow is assumed. The throughfall
(complemented with the amount of melted snow) gives the potential infiltration.
Important novelty of Biome-BGCMuSo v6.2 is that maximum infiltration is calculated
based on the saturated hydraulic conductivity and the SWC of the top soil layers. If the
potential infiltration exceeds the maximum infiltration, pond water can be formed. If the sum
of the precipitation and the actual pond height minus the maximum infiltration rate is greater
than the maximum pond height, the excess water is added to surface runoff detailed below
(Balsamo et al., 2009). The maximum pond height is an input parameter. Water from the pond
can infiltrate into the soil at a rate the top soil layer can absorb it. Evaporation of the pond
water is assumed equal to the potential evaporation.
Surface runoff is the water flow occurring on the surface when a portion of the
precipitation cannot infiltrate into the soil. Two types of surface runoff processes can be
distinguished: Hortonian and Dunne. Hortonian runoff is unsaturated overland flow that
occurs when the rate of precipitation exceeds the rate at which water can infiltrate. The other
type of surface runoff is the Dunne runoff (also known as the saturation overland flow) which
occurs when the entire soil is saturated but the rain continues to fall. In this case the rainfall
immediately triggers pond water formation and (above the maximum pond water height)



surface runoff. The handling of these processes is presented in the soil hydrological module of
Biome-BGCMuSo v6.2.
Calculation of Hortonain runoff (in kg $H_2O$ m$^{-2}$ day$^{-1}$) is based on a semi-empirical
method and uses the precipitation amount (in cm day$^{-1}$), the unitless runoff curve number
(*RCN*), and the actual moisture content status of the topsoil (Rawls et al., 1980; this method is
known as the SCS runoff curve number method). This type of runoff simulation can be turned
off by setting *RCN* to zero. The detailed description can be found in the Supplementary
material, Section 1. The amount of runoff as a function of the soil type and the actual SWC is
presented in Figure 1.

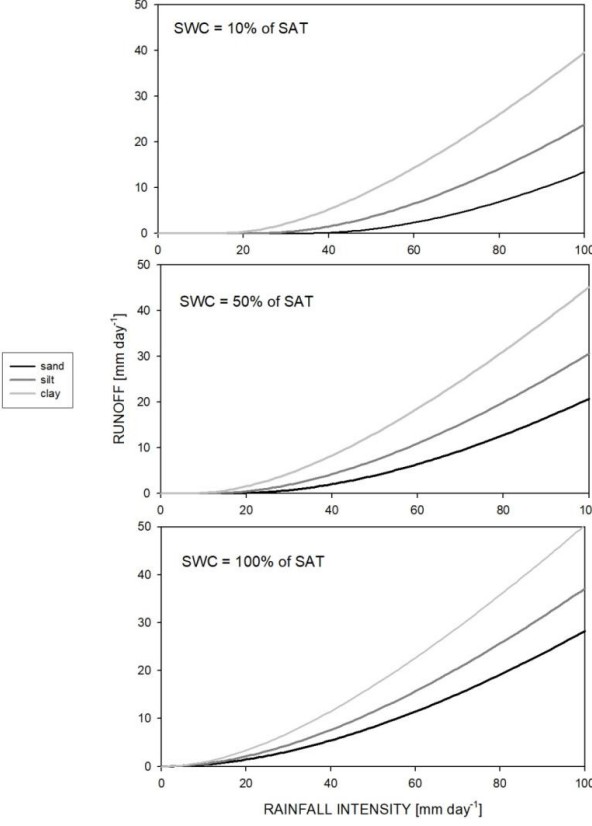


**Figure 1: Hortonian runoff as the function of rainfall intensity, soil type and actual soil water content of the top soil
layer. Sand soil means 92% sand, 4% silt and 4% clay; silt soil means 8% sand, 86% silt and 6% clay; clay soil means
20% sand, 20% silt and 60% clay. SWC means soil water content; SAT means saturation.**




### 3.2 Infiltration, percolation and diffusion

There are two optional methods in Biome-BGCMuSo v6.2 to calculate soil water movement between soil layers and actual SWC layer by layer. The first one is a cascade method (also known as tipping bucket method), and the second is a Richards equation based physical method. The tipping bucket method has a long history in crop modelling and is considered as a successful, well-evaluated algorithm that can accurately simulate downward water flow in the soil.

The cascade method uses a semi-empirical input parameter (*DC:* drainage coefficient in day$^{-1}$) to calculate downward water flow rate. When the SWC of a soil layer exceeds field capacity (FC), a fraction (equal to *DC*) of the water amount above FC goes to the layer next below. If *DC* is not set in the soil input file, it is estimated from the saturated hydraulic conductivity: $DC = 0.1122 \cdot K_{SAT}^{0.339}$ ($K_{SAT}$: saturated hydraulic conductivity in cm day$^{-1}$; the user can set its value or the model based on soil texture estimates it internally; see Hidy et al., 2016). The detailed description of the method can be found in the Supplementary material, Section 2. Drainage from the bottom layer is a net loss for the soil profile.

Water diffusion that is the capillary water flow between the soil layers is calculated to account for the relatively slow movement of water. The flow rate is the function of the water content difference of two adjacent layers and the soil water diffusivity at the boundary of the layers, which is determined based on the average water content of the two layers. The detailed mathematical description of the method can be found in the Supplementary material, Section 3.

The detailed description of the Richards method can be found in Hidy et al. (2012). To support efficient and robust calculations of soil water fluxes a dynamically changing time step was introduced in version 4.0 (Hidy et al., 2016). An enhancement of this method in Biome-BGCMuSo v6.2 is the finer vertical discretisation of the soil profile that is used during the numerical solution of the equation. The implementation of the Richards-equation is still in an experimental phase requiring rigorous testing and validation in the future.

### 3.3 Evapotranspiration

Biome-BGCMuSo, such as its predecessor Biome-BGC, estimates evaporation of leaf intercepted water, bare soil evaporation, and transpiration to estimate the total evapotranspiration in a daily level. The potential rates of all three processes are calculated based on the Penman-Monteith (PM) method. PM equation requires net radiation (minus soil


heat flux) and conductance values by definition using different parameterization for the different processes. The model calculates leaf- and canopy-level conductances of water vapour and sensible heat fluxes, to be used in Penman-Monteith calculations of canopy evaporation and canopy transpiration. Note that in the Biome-BGC model family the direct wind effect is ignored but can be considered indirectly by adjusting boundary layer conductance to site-specific conditions. A possible future direction might be the extension of the model logic to consider wind effect directly.

*3.3.1 Canopy evaporation*

If there is intercepted water, this portion of evaporation is calculated using the canopy resistance (reciprocal of conductance) to evaporated water and the resistance to sensible heat. The time required for the water to evaporate based on the average daily conditions is calculated, and subtracted from the day length to get the effective day length for evapotranspiration. Combined resistance to convective and radiative heat transfer is calculated based on canopy conductance of vapour and leaf conductance of sensible heat both of which are assumed to be equal to the boundary layer conductance. Besides the conductance/resistance parameters the canopy absorbed shortwave radiation drives the calculation. Note that the canopy evaporation routine was not modified significantly in Biome-BGCMuSo.

*3.3.2 Soil evaporation*

In order to estimate soil evaporation, first the potential evaporation is calculated, assuming that the resistance to vapour is equal to the resistance to sensible heat and assuming no additional resistance component. Both resistances are assumed to be equal to the actual aerodynamic resistance. Actual aerodynamic resistance is the function of the actual air pressure and air temperature and the potential aerodynamic resistance ($potR_{air}$ in s m$^{-1}$). $potR_{air}$ was a fixed value in the previous model versions (107 s m$^{-1}$). Its value was derived from observations over bare soil in tiger-bush in south-west Niger (Wallace and Holwill, 1997). In Biome-BGCMuSo v6.2, the $potR_{air}$ is an input parameter that can be adjusted by the user (Hidy et al., 2021). Another new development in Biome-BGCMuSo v6.2 is the introduction of an upper limit for daily potential evaporation ($evap_{limit}$) that is determined by the available energy (incident shortwave flux that reaches the soil surface):

$$evap_{limit} = \frac{irad \cdot dayl}{LH_{vap}} \tag{1}$$





where *irad* is the incident shortwave flux density in W m$^{-2}$, *dayl* is the length of the day in
seconds, $LH_{vap}$ is the latent heat of vaporization (the amount of energy that must be added to
liquid to transform into gas) in J kg$^{-1}$. This feature was missing from previous model versions
resulting in considerable overestimation of evaporation on certain days that was caused by the
missing energy limitation on evaporation.

An important novelty in Biome-BGCMuSo v6.2 is the calculation of the actual

evaporation from the potential evaporation and the square root of time elapsed since the last
precipitation (expressed by days; Ritchie, 1998). This is another method that has been used by
the crop modeller community for many years. Detailed description of the algorithm can be
found in the Supplementary material, Section 4.

A major novel feature in Biome-BGCMuSo v6.2 is the simulation of the reducing

effect of surface residue or mulch cover on bare soil evaporation. Here we use the term
'mulch' to quantify surface residue cover in general keeping in mind that mulch is typically a
human-induced coverage. Surface residue includes aboveground litter and coarse woody
debris as well.

The evaporation reduction effect (*evapREDmulch*; unitless) is a variable between 0

and 1 (0 means full limitation, and 1 means no limitation) estimated based on a power
function of the surface coverage (*mulchCOV* in %) and a soil specific constant set by the user
(*pREDmulch*; see Hidy et al., 2021). If variable *mulchCOV* reaches 100% it means that the
surface is completely covered. If *mulchCOV* is greater than 100% it means the surface is
covered by more than one layers. Surface coverage is a power function of the amount of
mulch (*mu* in kgC m$^{-2}$) with parameters $p1_{mulch}$, $p2_{mulch}$, and $p3_{mulch}$ (soil parameters) based
on the method of Rawls et al. (1991):

$$mulchCOV = p1_{mulch} \cdot (mu/p2_{mulch})^{p3_{mulch}} \tag{2}$$
$$evapREdmulch = pREDmulch^{\frac{mulchCOV}{100}} \tag{3}$$

Another simulated effect of surface residue cover is the homogenization of soil

temperature between 0 and 30 cm depth (layers 1, 2 and 3). The functional forms of surface
coverage and evaporation reduction factor are presented in Figure 2.

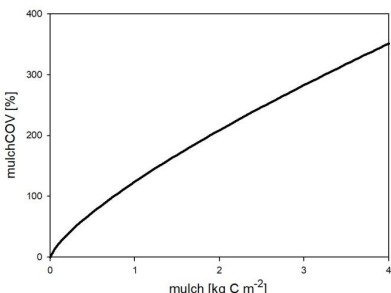

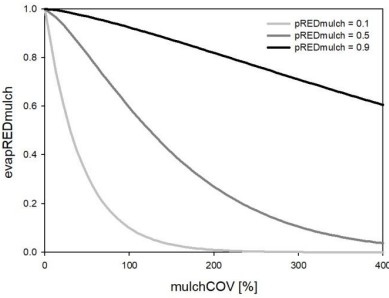


**Figure 2: Surface coverage as a function of the amount of surface residue or mulch (upper plot) and the evaporation reduction factor (evapREDmulch) as the function of mulch coverage (lower plot) using different mulch specific soil parameters (pREDmulch). See text for details.**


### 3.3.3 Transpiration

In order to simulate transpiration, first transpiration demand ($TD$ in kg $H_2O$ $m^{-2}$ $day^{-1}$) is calculated using the Penman-Monteith equation separately for sunlit and shaded leaves. $TD$ is the function of leaf-scale conductance to water vapor, which is derived from stomatal, cuticular and leaf boundary layer conductances. A novelty in Biome-BGCMuSo v6.2 is that potential evapotranspiration is also calculated using the maximal stomatal conductance instead of the actual stomatal conductance, which means that stomatal aperture is not affected by the soil moisture status (in contrast to the actual one).

$TD$ is distributed across the soil layers according to the actual root distribution using an improved method (the logic was changed since Biome-BGCMuSo v4.0). From the plant specific root parameters and the actual root weight Biome-BGCMuSo calculates the number of the layers where roots can be found together with the root mass distribution across the layers. If there is not enough water in a given soil layer to fulfil the transpiration demand, the transpiration flux from that layer is limited, and below wilting point (WP) it is set to zero. The sum of layer-specific transpiration fluxes across the root zone gives the actual transpiration

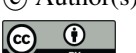



flux. The detailed description of the algorithm can be found in the Supplementary material,
Section 5.

### 3.4 Effect of groundwater

Simulation of groundwater effect was introduced in Biome-BGCMuSo v4.0 (Hidy et
al., 2016), but the method has been significantly improved, and the new algorithm it is now
available in Biome-BGCMuSo v6.2. In the recent model version there is an option to provide
an additional input file with the daily values of the groundwater table depth ($GWdepth$ in m).
Groundwater may affect soil hydrological and plant physiological processes if the
water table is closer to the root zone than the thickness of the capillary fringe (that is the
region saturated from groundwater via capillary effect). The thickness of the capillary fringe
($CF$ in m) is estimated using literature data and depends on the soil type (Johnson and Ettinger
model; Tillman and Weaver, 2006). Groundwater table distance ($GWdist$ in m) for a given
layer is defined as the difference between $GWdepth$ and the depth of the midpoint of the
layer.
The layers completely below the groundwater table are assumed to be fully saturated.
In case of layers within the capillary fringe ($GWdist < CF$), the calculation of water balance
changes: the field capacity rises, thus the difference between saturation (SAT) and FC
decreases and the layer charges gradually, till the increased FC value is reached. The FC-
rising effect of groundwater for the layers above the water table is calculated based on the
ratio of the groundwater distance and the capillary fringe thickness, but only after the water
content of the layers below have reached their modified FC values. Detailed description of the
groundwater effect can be found in Supplementary material, Section 6.

### 3.5 Soil moisture stress

In the original Biome-BGC model the effect of changing soil water content on
photosynthesis and decomposition of soil organic matter is expressed in terms of soil water
potential ($\Psi$). Instead of $\Psi$, the volumetric SWC is also widely used to calculate the limitation
of stomatal conductance and decomposition. A practical advantage of using SWC as a factor
in stress function is that it is easier to measure in the field and the changes of the driving
function are much smoother than in case of $\Psi$. The disadvantage is that SWC is not
comparable among different soil types (in contrast to $\Psi$).





The maximum of SWC is the saturation value; the minimum is the wilting point or the
hygroscopic water depending on the type of the simulated process. Novelty of Biome-
BGCMuSo v6.2 is that the hygroscopic water, the wilting point, the field capacity and the
saturation values are calculated internally by the model based on the soil texture data, or can
be defined in the input file layer by layer.
In Biome-BGCMuSo v6.2 the so-called soil moisture stress index (*SMSI*) is calculated
to represent overall soil stress conditions. *SMSI* is affected by the length of the drought event
(*SMSE:* extent of soil stress), the severity of the drought event (*SMSL:* length of soil stress*)*,
aggravated by the extreme temperature (*extremT:* effect of extreme heat). *SMSI* is equal to
zero if no soil moisture limitation occurs and equal to 1 in case of full soil moisture limitation.
*SMSI* is used by the model for plant senescence calculations (presentation of plant related
processes is the subject of a forthcoming publication)). The members of *SMSI* are explained
detailed below.

$SMSI = 1 - SMSE \cdot SMSL \cdot extremT$                              (4)
Magnitude of soil moisture stress (*SMSE*) is calculated layer by layer based on SWC.
Regarding soil moisture stress two different processes are distinguished: drought (i.e. low
SWC close to or below WP) and anoxic condition (i.e. after large precipitation events or in
the presence of high groundwater table; Bond-Lamberty et al., 2007). An important novelty of
Biome-BGCMuSo v6.2 is the soil curvature parameters (*q*) which is introduced to provide
mechanism for soil texture dependent drought stress as it can affect the shape of the soil stress
function (which means possibility for non-linear ramp function):

$SMSE^i = 0$                                         $; SWC^i < SWC_{WP}^i$
$SMSE^i = \left(\frac{SWC^i - SWC_{WP}^i}{SWC_{drought}^i - SWC_{WP}^i}\right)^q$       $; SWC_{WP}^i \leq SWC < SWC_{drought}^i$       (5)
$SMSE^i = 1$                                         $; SWC_{drought}^i \leq SWC \leq SWC_{anoxic}^i$
$SMSE^i = \frac{SWC_{SAT}^i - SWC^i}{SWC_{SAT}^i - SWC_{anoxic}^i}$       $; SWC^i > SWC_{anoxic}^i$
where $q$ is the curvature of soil stress function, $SWC_{drought}^i$ and $SWC_{anoxic}^i$ are critical SWC
values for calculating soil stress.
In order to make the SWC values comparable between different soil types,
$SWC_{drought}^i$ and $SWC_{anoxic}^i$ can be set in normalized form (such as in Eq. 4) ) as part of the





ecophysiological parameterization of the model. More details about the adjustment of the critical SWC values can be found in Hidy et al. (2021).

The layer specific soil moisture stress extent values are summed across the root zone using the relative amount of roots in the layers ($RP^i$) as weighting factors to obtain the overall soil moisture stress extent ($SMSE$):

$$SMSE = \sum_{i=0}^{i=nr} SMSE^i \cdot RP^i \tag{6}$$

$$RP^i = RD \frac{z^i}{RL} \cdot e^{-RD \cdot (mid^i/RL)} \tag{7}$$

where $nr$ is the number of the soil layers where roots can be found, $RL$ is the actual length of roots, $RD$ is rooting distribution parameter (ecophysiological parameter; see details in the User's Guide; Hidy et al., 2021). In the current model version $SMSE$ can also affect the entire photosynthetic machinery by the introduction of an empirical parameter. This mechanism is responsible to account for the non-stomatal effect of drought on photosynthesis (details about this algorithm will be published in a separate paper). Since there is no mechanistic representation behind this empirical down-regulation of photosynthesis, further test are needed for the correct setting of this parameter using preferentially eddy covariance data.

The soil moisture stress length related factor ($SMSL$) is the ratio of the critical soil moisture stress length (ecophysiological parameter) and the sum of the daily $(1 - SMSE)$ values. This cumulated value restarts if $SMSE$ is equal to one (no stress). Extreme heat ($extremT$) is also considered and is taken into account in the final stress function (see above) by using a ramp function. Its parameterization thus requires the setting of two critical temperature limits that defines the ramp function (set by the ecophysiological parameterization; see Hidy et al., 2021). Its characteristic temperature values can be set by parameterization (ecophysiological input file).

## 4. Soil carbon and nitrogen cycles

### 4.1 Soil-litter module

We made substantial changes in the soil biogeochemistry module of the Biome-BGC model. Previous model versions already offered solutions for multilayer simulations (Hidy et al., 2012, 2016), but some pools still inherited the single-layer logic of the original model. In the new model version all relevant soil processes are separated layer by layer which is a major step forward.



Instead of defining a single litter, soil organic carbon (SOC) and nitrogen pool, we implemented separate carbon and nitrogen pools for each soil layer in the form of soil organic matter (SOM) and litter in Biome-BGCMuSo v6.2. The changes of the mass of the carbon and nitrogen pools are calculated layer by layer. Mortality fluxes (whole plant mortality, senescence, litterfall) of aboveground plant material are transferred into the litter pools of the top soil layers (0-10 cm, layers 1-2). Mortality fluxes of belowground plant material are transferred into the corresponding soil layers based on their location within the root zone. Due to ploughing and leaching, carbon and nitrogen can also be relocated to deeper layers. The plant material turning into the litter compartment is divided between the different types of litter pools (labile, unshielded cellulose, shielded cellulose and lignin) according to the parameterization. Litter and soil decomposition fluxes (carbon and nitrogen fluxes from litter to soil pools) are calculated layer by layer, depending on the actual temperature and SWC of the corresponding layers. Vertical mixing of soil organic matter between the soil layers (e.g. bioturbation) is not implemented in the current model version.

Figure 3 shows the most important simulated soil and litter processes. N-fixation ($Nf$) is the N input from the atmosphere to soil layers in the root zone by microorganisms. The user can set its annual value as an input parameter. N-deposition ($Nd$) is the N input from the atmosphere to the top soil layers (see below). The user can set its annual value as a site-specific parameter in the initialization input file. Nitrogen deposition can be provided by annually varying values as well. Plant uptake ($PU$) is the absorption of mineral N by plants from the soil layers in the root zone. Mineralization ($MI$) is the release of plant-available nitrogen (flux from soil organic matter to mineralized nitrogen). Immobilization ($IM$) is the consumption of inorganic nitrogen by microorganisms (flux from mineralized nitrogen to soil organic matter). Nitrification ($NI$) is the biological oxidation of ammonium to nitrate through nitrifying bacteria. Denitrification ($DN$) is a microbial process where nitrate ($NO_3^-$) is reduced and converted to nitrogen gas ($N_2$) through intermediate nitrogen oxide gases. Leaching ($L$) is the loss of water-soluble mineral nitrogen from the soil layers. If leaching occurs in the lowermost soil layer that means loss of N from the simulated system. Litterfall ($LI$) is the plant material transfer from plant compartments to litter. Decomposition is the C and N transfer from litter to soil pools and between soil pools. In case of woody vegetation coarse woody debris (CWD) contains the woody plant material after litterfall before physical fragmentation. Litter has also four sub-pools based on their composition: labile (L1), unshielded and shielded cellulose (L2, L3) and lignin (L4). Soil organic matter has also four



sub-pools based on their turnover rate: labile (S1), medium (S2), slow (S3) and passive

(recalcitrant; S4) SOM pool. Soil mineralized nitrogen pool contains the inorganic N-forms of

the soil: ammonium and nitrate.

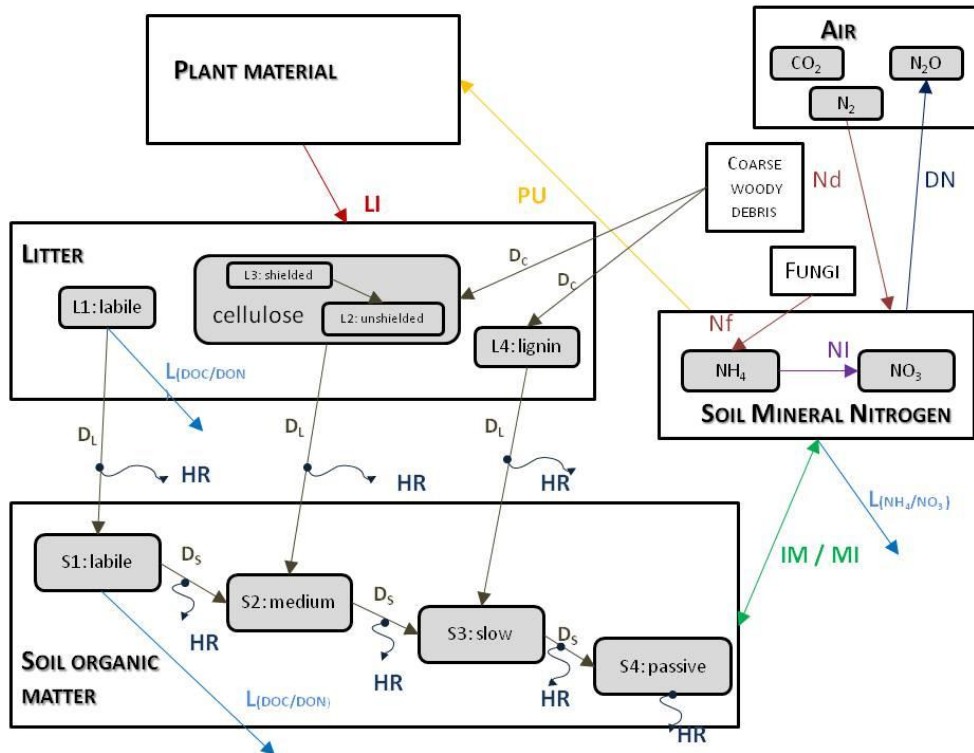

**Figure 3: Soil and litter related simulated carbon/nitrogen fluxes (arrows) and pools (rectangles) in Biome-BGCMuSo v6.2. HR: heterotrophic respiration, IM: immobilization, MI: mineralization, PU: plant uptake, LI: litterfall, NI: nitrification, D: decomposition ($D_L$: decomposition of litter, $D_S$: decomposition of SOM, $D_C$: fragmentation of coarse woody debris), L: leaching, Nf: nitrogen fixation, Nd: nitrogen deposition, DN: denitrification. L represents loss of C and N from the simulated system.**


**4.2 Decomposition**

In the decomposition module (i.e. converging cascade scheme; Thornton, 1998) the

fluxes between litter and soil pools are calculated layer by layer. The potential fluxes are
modified in case of N limitation when the potential gross immobilization is greater than the
potential gross mineralization.

To explain the decomposition processes implemented in Biome-BGCMuSo v6.2 the

main carbon/nitrogen pools and fluxes between litter and soil organic and inorganic
(mineralized) matter are presented on Figure 4.



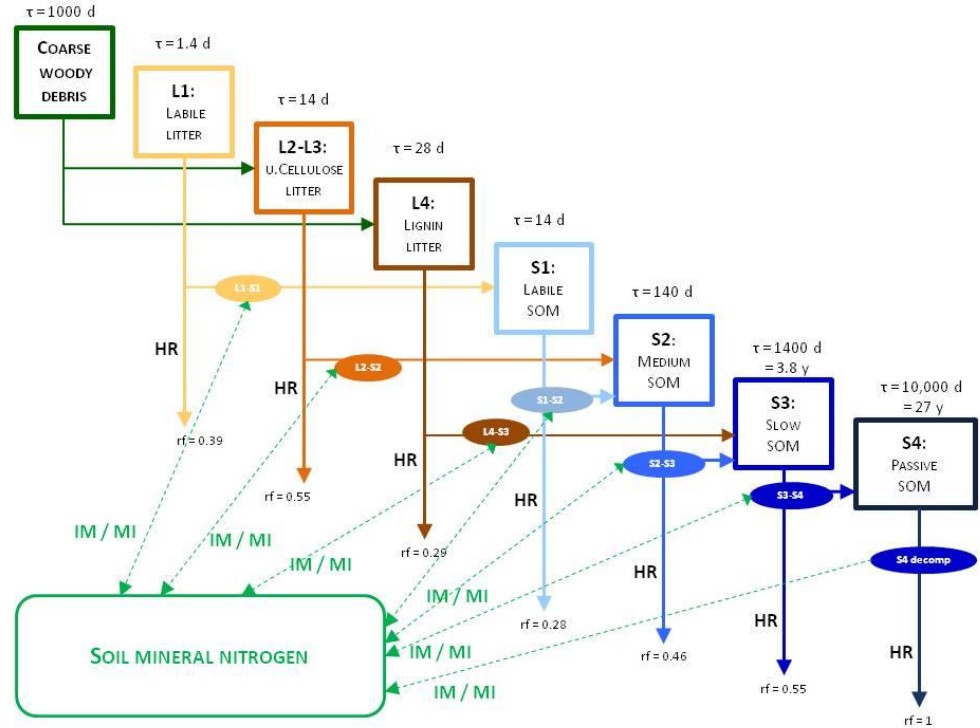


**Figure 4: Overview of the converging cascade model of litter and soil organic matter decomposition that is**
**implemented in Biome-BGCMuSo v6.2. rf represents the respiration fraction of the different transformation fluxes, τ**
**is the residence time (reciprocal of the rate constants that is the turnover rate), IM/MI: immobilization/mineralization**
**fluxes, HR: heterotrophic respiration. Note that both the respiration fraction and the turnover rate parameters can be**
**adjusted through parameterization.**


For the calculation of nitrogen mineralization first respiration cost (respiration
fraction) is estimated. Mineralization than is the function of the remaining part of the pool and
its C:N ratio. The nitrogen mineralization fluxes of the SOM pools are functions of the
potential rate constant (reciprocal of residence time), and the integrated response function that
accounts for the impact of multiple environmental factors. The integrated response function of
decomposition is a product of the response functions of depth, soil temperature and SWC
($F_r(d)_D$, $F_r(T)_D$, $F_r(SWC)_D$; Figure 5). Its detailed description can be found in the
Supplementary material, Section 7. The dependence of the three different factors on depth,
temperature and SWC with default parameters are presented in Figure 5.

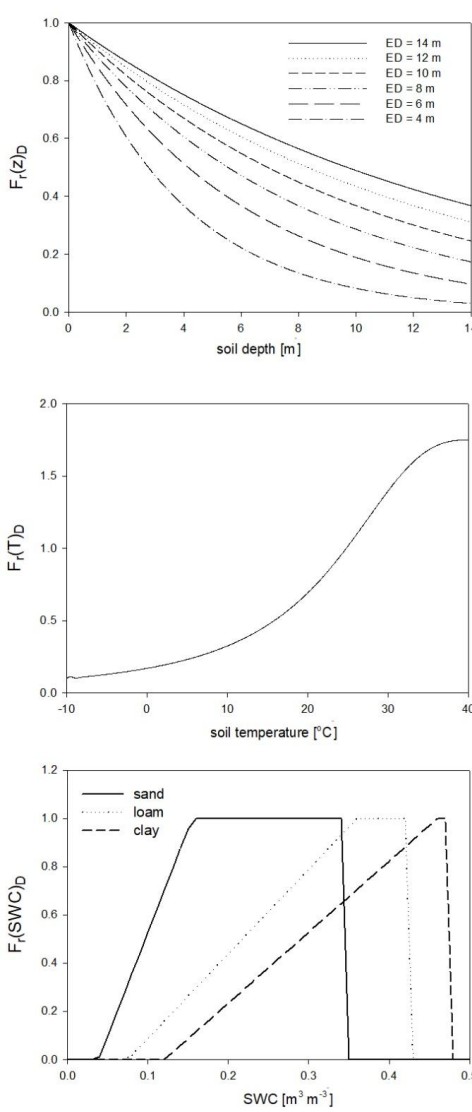


**Figure 5: The dependence of the individual factors that form the complex environmental response function of decomposition on depth (Fr(d)$_D$), temperature (Fr(T)$_D$) and SWC in case of different soil types (Fr(SWC)$_D$). ED is the e-folding depth which is one of the adjustable soil parameters of the model. Sand soil means 92% sand, 4% silt and 4% clay; silt soil means 8% sand, 86% silt and 6% clay; clay soil means 20% sand, 20% silt and 60% clay.**


### 4.3 Soil nitrogen processes

In Biome-BGCMuSo v6.2 separate ammonium (*sNH4*) and nitrate (*sNO3*) soil pools
are implemented instead of a general mineralized nitrogen pool. This was a necessary step for



the realistic representation of many internal processes like plant nitrogen uptake, nitrification,
denitrification, consideration of the effect of different mineral and organic fertilizers and $N_2O$
emission.
It is important to introduce the *availability* concept that Biome-BGCMuSo uses and is
associated with the ammonium and nitrate pools. We use the logic proposed by Thomas et al.
(2013) which means that the plant has access only to a part of the given inorganic nitrogen
pool. Unavailable part is buffered as it is associated with soil aggregates and is unavailable for
plant uptake. The available part of ammonium is calculated based on *NH₄ mobilen proportion*
(that is a soil parameter set to 10% according to Thomas et al., 2013; Hidy et al., 2021) and
the actual pool. The available part of nitrate is assumed to be 100%.
The amount of ammonium and nitrate are determined layer by layer controlled by
input and output fluxes (*F* in kg N m⁻² day⁻¹) listed below:
$F_{sNH4}^i = IN_{sNH4}^i - L_{sNH4}^i + L_{sNH4}^{i-1} - PU_{sNH4}^i - IM_{sNH4}^i + MI_{sNH4}^i - NI_{sNH4}^i$ (8)
$F_{sNO3}^i = IN_{sNO3}^i - L_{sNO3}^i + L_{sNO3}^{i-1} - PU_{sNO3}^i - IM_{sNO3}^i + MI_{sNO3}^i - DN_{sNO3}^i$ (9)
where $IN_{sNH4}^i$ and $IN_{sNO3}^i$ are the input fluxes to the ammonium and nitrate pools, respectively;
$L_{sNH4}^i$ , $L_{sNH4}^{i-1}$, $L_{sNO3}^i$, $L_{sNO3}^{i-1}$ are the amount of leached mineralized ammonium and nitrate from
a layer (*i*) or from the upper layer (*i-1*), respectively; $PU_{sNH4}^i$ and $PU_{sNO3}^i$ are the plant uptake
fluxes of ammonium and nitrate, respectively; $IM_{sNH4}^i$ and $IM_{sNO3}^i$ are the immobilization
fluxes of ammonium and nitrate, respectively; $MI_{sNH4}^i$ and $MI_{sNO3}^i$ are the mineralization
fluxes of ammonium and nitrate, respectively; $NI_{sNH4}^i$ is the nitrification flux of ammonium
and $DN_{sNO3}^i$ is the denitrification flux of nitrate.
In the following subsections the different terms of the equations are described in
detail.
Input to the sNH4 and sNO3 pools (IN in Eq. 6 and 7)
According to the model logic N-fixation occurs in the root zone layers. Its distribution
between sNH4 and sNO3 pools is calculated based on their actual available proportion in the
actual layer ($NH4prop^i$):
$NH4prop^i = sNH4avail^i + sNO3avail^i$ (10)
where $sNH4avail^i$ and $sNO3avail^i$ are the available part of the sNH4 and sNO3 pools in the
actual layer.
N-deposition related nitrogen input is associated with the 0-10 cm soil layers assuming
uniform distribution across layers 1-2 in the model, and the distribution between sNH4 and





sNO3 pools is calculated based on the *proportion of NH₄ flux of N-deposition* soil parameter
(Hidy et al., 2021).

Organic and inorganic fertilization is also an optional nitrogen input. The amount and

composition ($NH_4^+$ and $NO_3^-$ content) can be set in the fertilization input file.

Leaching - downward movement of mineralized N (L in Eq. 6 and 7)

The amount of leached mineralized N (mobile part of the given N pool) from a layer is

directly proportional to the amount of drainage and the available part of the sNH4 and sNO3
pools. Leaching from the layer above is a net gain, while leaching from actual layer is a net
loss for the actual layer. Leaching is described in Section 4.5.

Plant uptake by roots (PU in Eq. 6 and 7)

N uptake required for plant growth is estimated in the photosynthesis calculations and

the amount is distributed across the layers in the root zone. The partition of the N uptake
between sNH4 and sNO3 pools is calculated based on their actual available proportion in each
layer.

Mineralization and immobilization (MI and IM Eq. 6 and 7)

Mineralization and immobilization calculations are detailed in Section 4.2. The

distribution of these N fluxes between sNH4 and sNO3 pools is calculated based on their
actual available proportion in each layer.

Nitrification (NI Eq. 6 and 7)

Nitrification is a function of the soil ammonium content, the net mineralization and the

response functions of temperature, soil pH and SWC ($F_r(pH)_{NI}$, $F_r(T)_{NI}$, and $F_r(SWC)_{NI}$,
respectively) based on the method of Parton et al. (2001) and Thomas et al. (2013). Its
detailed mathematical description can be found in the Supplementary material, Section 8. The
response functions with proposed parameters are shown in Figure 6.



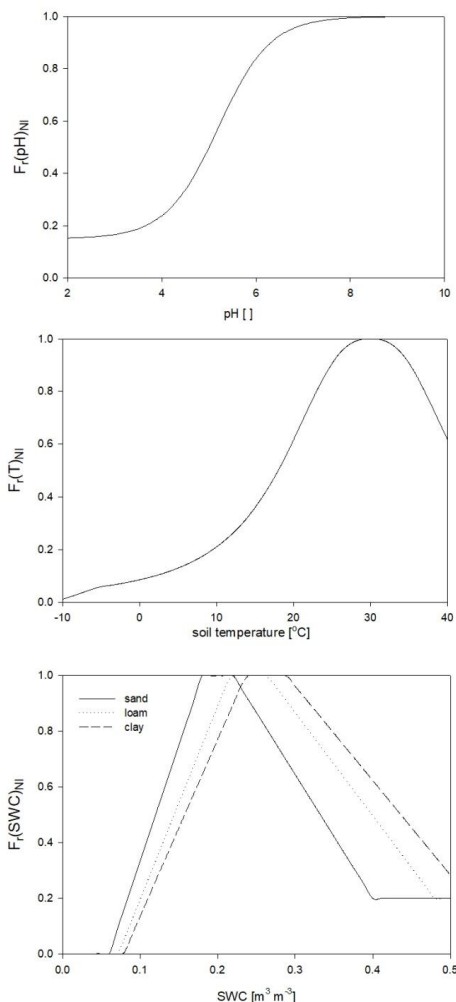

**Figure 6: The dependence of the individual factors of the environmental response function of nitrification on soil pH ($F_r(pH)_{NI}$), temperature ($F_r(T)_{NI}$) and SWC $F_r(SWC)_{NI}$ in case of different soil types. pH and temperature response functions are independent of the soil texture.**

Denitrification (DN Eq. 6 and 7)

Denitrification flux is estimated with a simple formula (Thomas et al., 2013):

$$DN^i = DNcoeff \cdot SOMresp^i \cdot sNO3avail^i \cdot WFPS^i \qquad (11)$$

where $DN$ of the actual layer is the product of the available nitrate content ($sNO3avail$ in kg N m$^{-2}$), $SOMresp^i$ in g C m$^{-2}$ day$^{-1}$ is the SOM decomposition related respiration cost, the $WFPS^i$ is the water-filled pore space and $DNcoeff$ is the *soil respiration related denitrification rate* in g C$^{-1}$, which is an input soil parameter (Hidy et al., 2021). The unitless



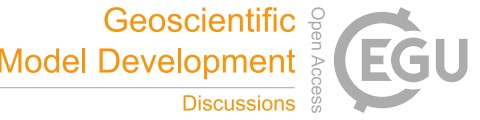

water-filled pore space is the ratio of the actual and the saturated SWC. SOM decomposition
associated respiration is the sum of the heterotrophic respiration fluxes of the four soil
compartments (S1-S4, Figure 4.).

**4.4 $N_2O$-emission and N-emission**

During both nitrification and denitrification $N_2O$-emission occurs which (added to the
$N_2O$-flux originated from grazing processes if applicable) contributes to the total $N_2O$-
emission of the examined ecosystem.
In Biome-BGCMuSo v6.2 a fixed part (set by the *coefficient of $N_2O$ emission of*
*nitrification* input soil parameter; Hidy et al., 2021) of nitrification flux is lost as $N_2O$ and not
converted to $NO_3$.
During denitrification, nitrate is transformed into $N_2$ and $N_2O$ gas depending on the
environmental conditions: $NO_3$ availability, total soil respiration (proxy for microbial
activity), SWC and pH. The *denitrification related $N_2/N_2O$ ratio* input soil parameter is used
to represent the effect of the soil type on the $N_2/N_2O$ ratio (del Grosso et al., 2000; Hidy et al.,
2021). Detailed mathematical description of the algorithm can be found in the Supplementary
material, Section 9.

**4.5 Leaching of dissolved matter**
Leaching of nitrate, ammonium, and dissolved organic carbon and nitrogen (DOC and
DON) content from the actual layer is calculated as the product of the concentration of the
dissolved component in the soil water and the amount of water (drainage plus diffusion)
leaving the given layer either downward or upward. The dissolved component (concentration)
of organic carbon is calculated from the SOC pool contents and the corresponding *fraction of*
*dissolved part of SOC* soil parameters. The dissolved component of organic nitrogen content
of the given soil pool is calculated from the carbon content and the corresponding C:N ratio.
The downward leaching is net loss from the actual layer and net gain for the layer next below;
the upward flux is net loss for the actual layer and net gain for the layer next up. The
downward leaching of the bottom active layer ($9^{th}$) is net loss for the system. The upward
movement of dissolved substance from the passive ($10^{th}$) layer is net gain for the system.




## 5. Case studies

### 5.1 Evaluation of soil hydrological simulation

In order to evaluate the functioning of the new model version (and to compare simulation results made by the current and the previously published model version), a case study is presented regarding soil water content and soil evaporation simulations. The results of a bare soil simulation (i.e. no plant is assumed to be present) are compared to observation data of a weighing lysimeter station installed at Martonvásár, Hungary (47°18'57.6"N, 18°47'25.6"E) in 2017. The station consists of twelve 2 meter deep scientific lysimeter columns with 1 m diameter (Meter Group Inc., USA) with soil temperature, SWC and soil water potential sensors installed at 5, 10, 30, 50, 70, 100 and 150 cm depth. Observation data for 2020 from six columns without vegetation cover (i.e. bare soil) was used to validate the model.

Raw lysimeter observation data were processed using standard methods. Bare soil evaporation values were derived based on changes of the mass of the soil columns also considering the mass change of the drainage water. Additionally, experience has shown that wind speed is related to the high frequency mass change of the soil column mass. To reduce noise, 5-point (5-min) moving averages were used based on Marek et al. (2014). After quality control of the data, the corrected and smoothed lysimeter mass values were used for the calculations. SWC observations were averaged to daily resolution to match the time step of the model.

Observed local meteorology was used to drive the models for year 2020. Soil physical model input parameters (field capacity, wilting point, bulk density, etc.) were determined in the laboratory using 100 cm$^3$ undisturbed soil samples taken from various depths during the installation of the lysimeter station. Regarding other soil parameters the proposed values were used. Detailed description of the input soil parameters and their proposed values are presented in the User's Guide (Hidy et al., 2021).

In Figure 7 the simulated and the observed time series of soil evaporation are presented for Martonvásár, for 2020. The figure shows that the soil evaporation simulation by v6.2 is more realistic than by v4.0. Biome-BGCMuSo v4.0 provides very low values during summer in some days which is not in accordance with the observations. Biome-BGCMuSo v6.2 provides more realistic values during this time period.





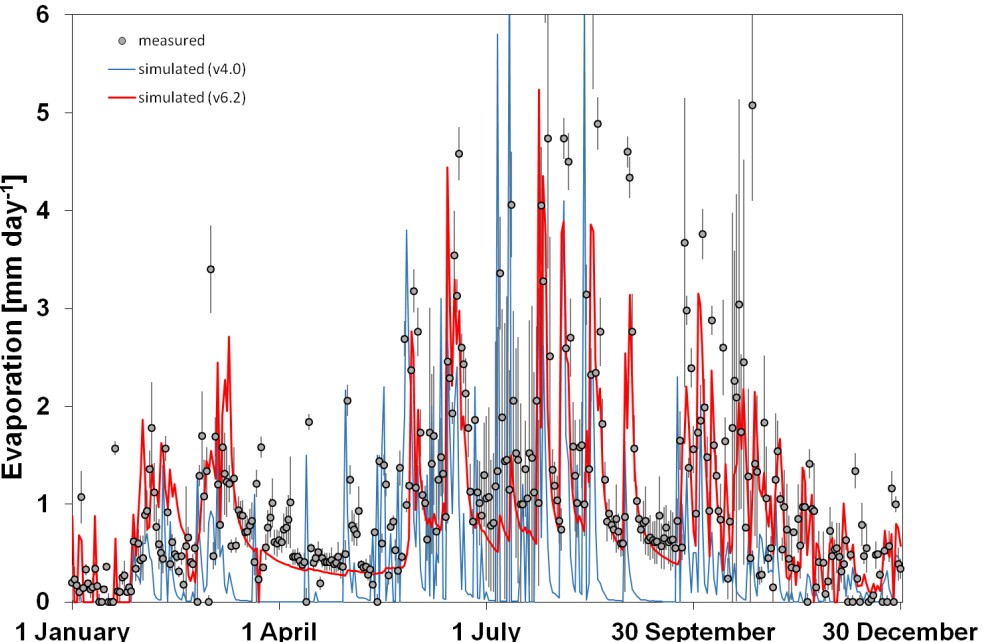

**Figure 7: The simulated (blue line: v4.0; red line: v6.2) and the observed (grey dots) daily soil evaporation values at Martonvásár during 2020. Vertical grey lines associated with the observations represent standard deviation of the observation from 6 columns. The improved model clearly outperforms the earlier version.**

In Figure 8 the simulated and the observed SWC at 10 cm depth are presented with the daily sum of precipitation representing the bare soil simulation in Martonvásár, for 2020. The soil water balance simulation seems to be realistic using v6.2, since the annual course captures the low and high end of the observed values. In contrast, Biome-BGCMuSo v4.0 underestimates the range of SWC and provides overestimations during the growing season (from spring to autumn). With a couple of exceptions, the simulated values using v6.2 fall into the uncertainty range of the measured values defined by the standard deviation of the six parallel measurements. This is not the case for the simulations with the 4.0 version.



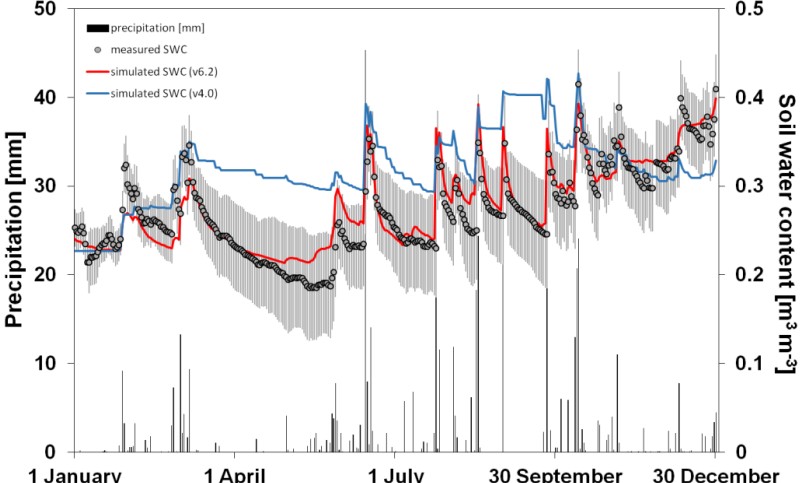

**Figure 8: The simulated (blue line: v4.0; red line: v6.2) and the observed (gray dots) soil water content values at 10 cm**
**depth (right y axis) with the daily sums of precipitation (left axis; black columns) during 2020 at Martonvásár**
**lysimeter station. Vertical grey lines associated with the observations represent +/- one standard deviation of the**
**observation. Simulated SWC using v6.2 is more consistent with the observations than using v4.0.**

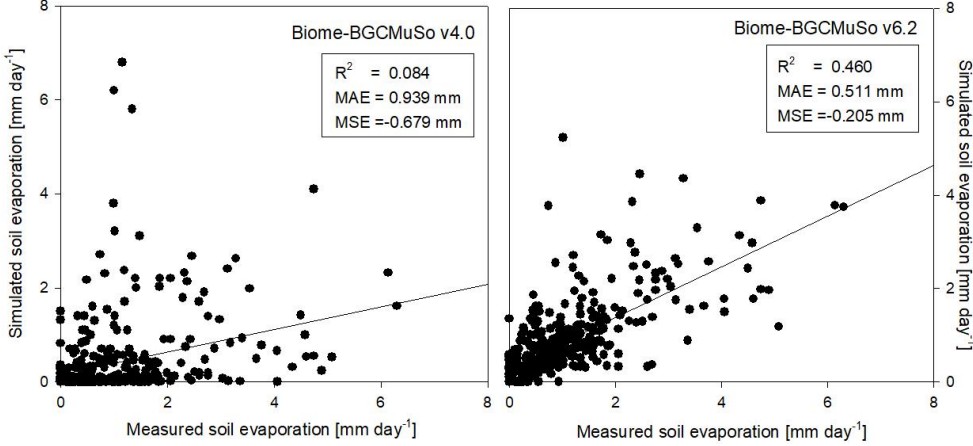

**Figure 9: Comparison of the simulated (left: v4.0; right: v6.2) and observed daily soil evaporation (right) representing**
**the means of measured data obtained from six weighing lysimeter columns with bare soil at Martonvásár in 2020. $R^2$,**
**MAE and MSE denote the square of the linear correlation coefficient, mean absolute error and mean signed error**
**(bias) of the simulated values, respectively.**



Model performance was evaluated by quantitative measures such as square of linear

correlation coefficient ($R^2$), mean absolute error (MAE) and mean signed error (MSE). In

Figure 9 the comparison of the simulated and the observed daily evaporation is presented.



Based on the performance indicators it is obvious that the simulation with new model version
(v6.2) is much closer to observations than the old version (v4.0). Biome-BGCMuSo v6.2
slightly underestimated the observations.
In Figure 10 the comparison of the simulated and the observed daily SWC from the
lysimeter experiment is presented. Based on the model evaluation it seems that the simulation
with new model version is much closer to observation than with old version (4.0). The results
obtained from v4.2 are consistent with earlier findings about the incorrect representation of
the annual SWC cycle (Hidy et al., 2016; Sándor et al., 2017).
Throughout validation of the improved model based on observed SWC and ET
datasets from eddy covariance sites is under way.

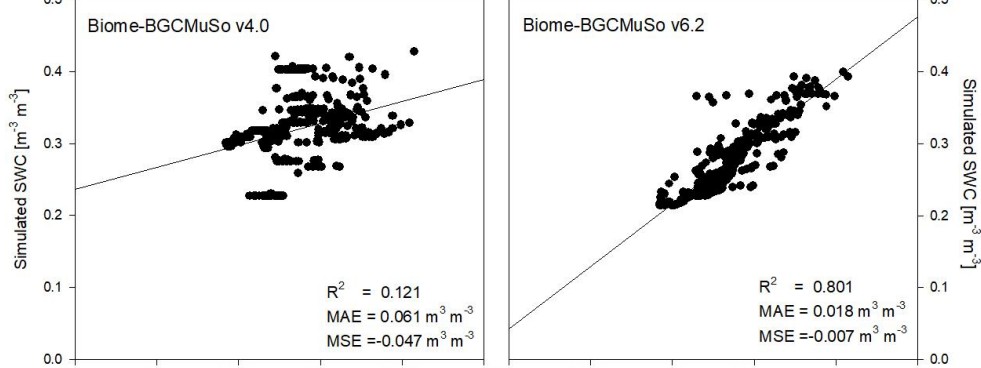

**Figure 10: Comparison of the simulated (left: v4.0; right: v6.2) and observed daily SWC representing the means of**
**measured data obtained from six weighing lysimeter columns with bare soil at Martonvásár in 2020. $R^2$, MAE and**
**MSE denote the square of the linear correlation coefficient, mean absolute error and mean signed error (bias) of the**
**simulated values, respectively.**

**5.2 Sensitivity analysis and optimization of the soil biogeochemistry scheme**

Here we present another case study that provides insight into the functioning of the
converging cascade scheme that is implemented in Biome-BGCMuSo v6.2. A large scale
experiment is also presented where the main aim was to perform model self-initialization (i.e.
spinup) at the country scale (for the entire area of Hungary) where the resulting soil organic
matter pools are expected to be consistent with the observations.



The observation based, gridded, multi-layer SOC database of Hungary (DOSoReMI
database; Pásztor et al., 2020) as well as the FORESEE meteorological database (Kern et al.,
2016) was used for the sensitivity analysis of the soil scheme as well as for optimizing the
most important soil parameters referring to SOC simulation. As a first step, the area of the
country was divided into 1104 grid cells (regular grid with 0.1° by 0.1° resolution). The 1104
grid cells of the DOSoReMI database were grouped based on their dominant land-use type
(cropland, grassland, forest based on CORINE-2012 database; EEA, 2021) as well as the soil
texture class (12 classes according to the USDA system; USDA, 1987) and SOC content (high
and low; high is greater than the group mean while low is less than the mean) of the topsoil
(0-30 cm layer). As some of the theoretically possible 72 groups had no members (e.g. there is
no soil in Hungary with sandy-clay texture) soils of the 1104 grid cells were categorized into
51 groups. For each group one single cell (so-called representative cell) was selected based on
the topsoil SOC content. The representative cell was the one with the smallest absolute
deviation from the group mean SOC content.
Grassland ecophysiological parameterization without management was used for
croplands for the spinup phase, and with fertilization, harvest and ploughing settings in the
transient phase. In case of grasslands, both during the spinup and transient phases grassland
parameterization was used, and in the transient phase mowing was assumed (once a year in
case of forests generic deciduous broadleaf forest parameterization was used for both spinup
and transient phases, and in the transient phase thinning was set. Parameterization was
performed based on generic, plant functional type specific ecophysiological parameters that
were created based on the original parameterization of White et al. (2000). Biome-BGCMuSo
specific parameter sets are available at the website of the model[1].
Soil parameters in Biome-BGCMuSo 6.2 were classified into six groups: (1) 4 generic
soil parameters, (2) 24 decomposition-nitrification-denitrification related parameters, (3) 14
rate scalars for the converging cascade scheme, (4) 19 soil moisture related parameters, (5) 7
methane related parameters and (6) 11 soil composition and characteristic values (can be set
layer by layer). Detailed description and proposed value of each soil parameters can be found
in the User's Guide (Hidy et al., 2021).


---

[1] http://nimbus.elte.hu/bbgc/files/generic_EPC_set_6.1.zip





**Table 1: Soil parameters of Biome-BGCMuSo v6.2 (referring to SOC simulation) that were used during the sensitivity**
**analysis. The first column contains the group, the second contains the name of the parameter, the third contains the**
**abbreviation, and the fourth contains original proposed values (Hidy et al., 2021). See Figure 4 for explanation on the**
**compartment names. The parameters that were included in the 2nd phase of the sensitivity analysis are marked with**
**bold letters (see text).**

| GROUP | NAME | ABBREVIATION | VALUE |
|---|---|---|---|
| Generic soil parameters | **C:N ratio of stable soil pool (soil4)** | **soil4CN** | **12** |
| | NH4 mobilen proportion | amMP | 0.1 |
| | aerodynamic resistance | potRair | 107 |
| Decomposition, nitrification, denitrification parameters | **parameter 1 for temperature response function of decomp.** | **Tp1decomp** | **1.75** |
| | **parameter 2 for temperature response function of decomp.** | **Tp2decomp** | **17** |
| | **parameter 3 for temperature response function of decomp.** | **Tp3decomp** | **2.6** |
| | **parameter 4 for temperature response function of decomp.** | **Tp4decomp** | **40** |
| | **minimum T for decomposition and nitrification** | **Tp5decomp** | **-5** |
| | e-folding depth of decomposition rate's depth scalar | EFD | 10 |
| | net mineralization proportion of nitrification | NITRnetMINER | 0.2 |
| | maximum nitrification rate | NITRmaxRATE | 0.1 |
| | coefficient of N2O emission of nitrification | NITRratioN2O | 0.02 |
| | parameter 1 for pH response function of nitrification | pHp1nitrif | 0.15 |
| | parameter 2 for pH response function of nitrification | pHp2nitrif | 1 |
| | parameter 3 for pH response function of nitrification | pHp3nitrif | 5.2 |
| | parameter 4 for pH response function of nitrification | pHp4nitrif | 0.55 |
| | parameter 1 for Tsoil response function of nitrification | Tp1nitrif | 1 |
| | parameter 2 for Tsoil response function of nitrification | Tp2nitrif | 12 |
| | parameter 3 for Tsoil response function of nitrification | Tp3nitrif | 2.6 |
| | parameter 4 for Tsoil response function of nitrification | Tp4nitrif | 2.6 |
| | minimum WFPS for scalar of nitrification calculation | minWFPS | 0.1 |
| | lower optimum WFPS for scalar of nitrification | opt1WFPS | 0.45 |
| | higher optimum WFPS for scalar of nitrification | opt2WFPS | 0.55 |
| | minimum value for saturated WFPS scalar of nitrification | minWFPSscalar | 0.2 |
| | **soil respiration related denitrification rate** | **DENITcoeff** | **0.05** |
| | denitrification related N2/N2O ratio multiplier | DNratioN2O | 2 |
| | **critical WFPS value for denitrification** | **critWFPSdenitr** | **0.50** |
| Rate scalars | **respiration fractions for fluxes between compartments (l1s1)** | **RFl1s1** | **0.39** |
| | **respiration fractions for fluxes between compartments (l2s2)** | **RFl2s2** | **0.55** |
| | **respiration fractions for fluxes between compartments (l4s3)** | **RFl4s3** | **0.29** |
| | **respiration fractions for fluxes between compartments (s1s2)** | **RFs1s2** | **0.28** |
| | **respiration fractions for fluxes between compartments (s2s3)** | **RFs2s3** | **0.46** |
| | **respiration fractions for fluxes between compartments (s3s4)** | **RFs3s4** | **0.55** |
| | potential rate constant of labile litter pool | RCS1 | 0.7 |
| | potential rate constant of cellulose litter pool | RCS2 | 0.07 |
| | potential rate constant of lignin litter pool | RCS3 | 0.014 |
| | potential rate constant of fast microbial recycling pool | RCS4 | 0.07 |
| | potential rate constant of medium microbial recycling pool | RCS5 | 0.014 |
| | **potential rate constant of slow microbial recycling pool** | **RCS6** | **0.0014** |
| | **potential rate constant of recalcitrant SOM (humus) pool** | **RCS7** | **0.0001** |
| | potential rate constant of physical fragmentation of wood | RCS8 | 0.001 |
| | maximum height of pond water | MP | 5 |
| | **curvature of soil stress function** | **q** | **1** |
| | fraction of dissolved part of S1 organic matter | fD1 | 0.005 |
| | fraction of dissolved part of S2 organic matter | fD2 | 0.004 |
| | **fraction of dissolved part of S3 organic matter** | **fD3** | **0.003** |
| | **fraction of dissolved part of S4 organic matter** | **fD4** | **0.002** |
| | mulch parameter: critical amount | CAmulch | 1 |
| | parameter 1 for mulch function | p1mulch | 100 |
| | parameter 2 for mulch function | p2mulch | 0.75 |
| | parameter 3 for mulch function | p3mulch | 0.75 |
| | mulch parameter: evaporation reduction | ERmulch | 0.5 |





As methane simulation was not the subject of the present case study we neglected the related parameters. Regarding to the soil composition and characteristic values we used the DOSoReMI database (Pásztor et al., 2020). From the remaining 61 parameters soil depth, runoff curve number, the three soil moisture related parameters (tipping bucket method) were not included into analysis. Groundwater parameters were inactive in this case (no groundwater is assumed) which means that those parameters were not studied. The remaining 53 parameters are used in sensitivity analysis and are listed in Table 1.

As a first step sensitivity analysis was carried out for the selected 53 soil parameters by running the Biome-BGCMuSo v6.2 model in spinup mode until a quasi-equilibrium in the total SOC is reached (that is the usual logic of the spinup run). The model was run for each representative cell 2000 times with varying model parameters using Monte-Carlo method. Each model parameters were varied randomly within the ±10% range of their initial values that were inherited from the Biome-BGC model or were set according to the literature. The least square linearization (LSL) method (Verbeeck et al., 2006) was used for dividing output uncertainty into its input parameter related variability. As result of the LSL method, the total variance of the model output and the sensitivity coefficient of each parameter can be determined. Sensitivity coefficients show the percent of total variance for which the given parameter is responsible.

In order to simplify the workflow and decrease the degree of freedom another sensitivity analysis was performed. In this second step, the sensitive parameters (sensitivity coefficient > 1% for at least one land use type; a total of 18 parameters) were used in the following sensitivity analysis with 6000 iteration steps. These 18 parameters are marked with bold letters in Table 1.

Figure 10 shows the summary of the second sensitivity analysis where the overall importance of the parameters are calculated as the mean of all selected pixels in a given land use category. It can be seen in Figure 10 that from the 18 parameters (selected during the first phase) soil carbon ratio of the recalcitrant pool (soil4CN), the temperature dependence parameters of decomposition function (Tp1decomp, Tp2decomp, Tp3decomp, Tp4_decomp) and the respiration fraction of S2-S3 and S3-S4 decomposition process (RFs2s3 and RFs3s4), the curvature of soil stress function ($q_{soilstress}$) and the fraction of dissolved part of S4 organic matter (fD4) are the most important for all land use types. Among the other parameters the critical WFPS of denitrification (critWFPSdentir) for grasslands has a remarkably high sensitivity (greater that 35%). It means that in case of grasslands the nitrogen availability seems to be an important limitation of the primary production, probably because there are



only natural sources of nitrogen (no fertilization is assumed here), and the rooting zone is
shallower than in case of forest which involves limited mineralized N access. Thus, in case of
higher values of critical WFPS of denitrification, the simulated production of the grassland
(and therefore the final SOC) seems to be significantly underestimated.

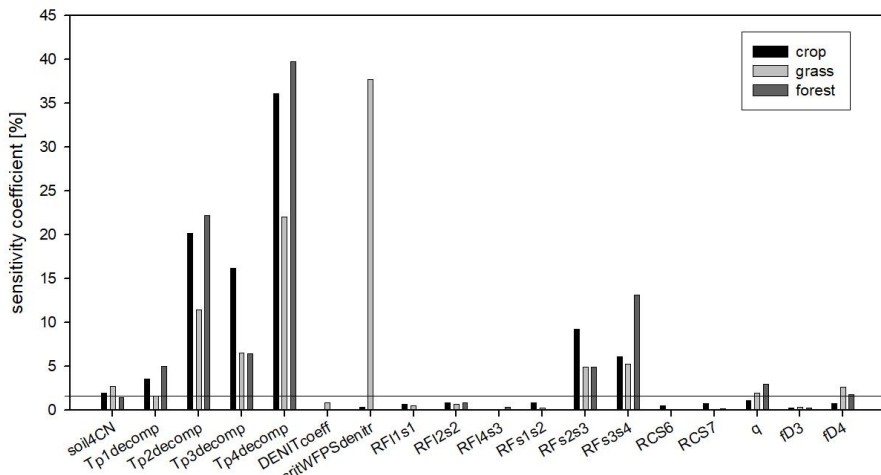

**Figure 10: The sensitivity coefficients of the soil parameters as the result of the sensitivity analysis. Black columns**
**refer to the crop, light grey to the grass and dark grey to the forest simulations. The sensitivity coefficients are**
**calculated as the mean pixel level sensitivity coefficient for the given land use type. Horizontal line indicates the 5%**
**threshold that was used to select the final parameter set that is subject to optimization.**

The selected ten, soil biogeochemistry related parameters were optimized for each of
the 51 groups separately, using maximum likelihood estimation. For each group, the
parameter set providing the smallest deviation between the simulated and the observed values
of the weighted average SOC content (weight factor of 5 is used for the 0-30 cm, and weight
factor of 1 is used for the 30-60 cm soil layers) was considered to be the final (optimized)
model parameter set.
The differences of the simulated and observed SOC content for the 0-30 cm layer
(SOC0-30) using the initial (Table 1) and final soil parameters (not shown here) are presented
in Figure 11. On the upper plot the signed relative error of SOC0-30 simulation before
optimization, while on the lower figure the signed relative error of SOC0-30 simulation after
optimization can be seen. It is clearly visible that because of optimization the overestimation
of the SOC0-30 simulation significantly decreased.





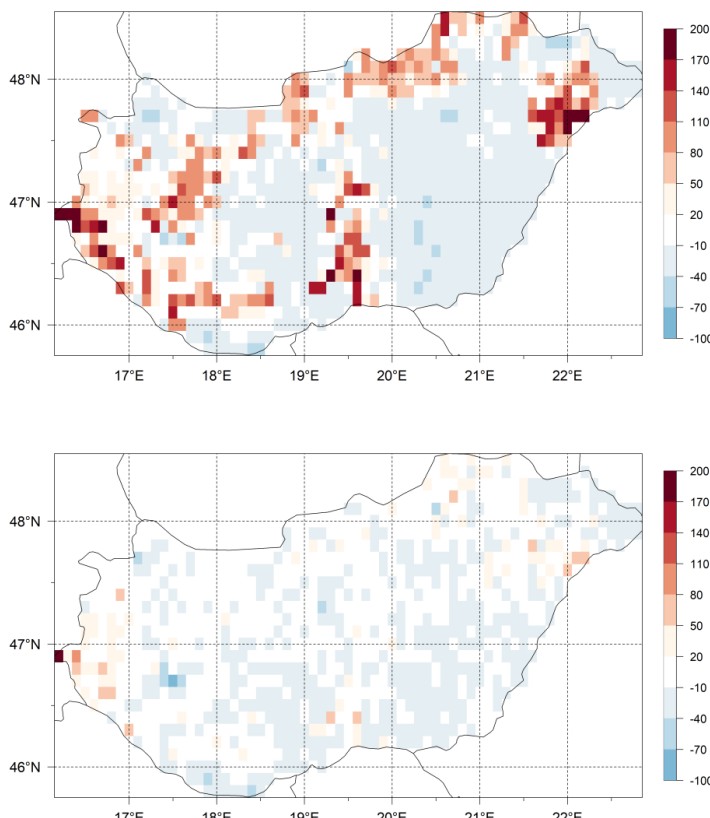

**Figure 11: Differences between the simulated and observed SOC data for the 0-30 layer (SOC0-30) using the initial (upper map) and optimized (lower map) soil parameters. The maps present the signed relative error in percent. Visual comparison of the maps reveals the success of the optimization in terms of capturing the overall SOC for the country area.**

We do not claim of course that the optimized parameters have universal value. Site history is neglected during the spin up simulations, and we use many simplifications like non-existent land use change, present-day ecosphysiological parameterization etc. In this sense, the optimized parameter set can be best considered as a pragmatic solution to provide initial conditions (equilibrium SOC pools) for the model at the country scale that is consistent with the observations.

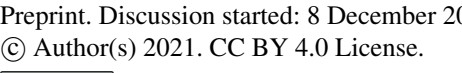





## 6. Concluding remarks


In this paper, we presented a detailed description of the soil hydrology and
carbon/nitrogen budget related developments of the Biome-BGCMuSo v6.2 terrestrial
ecosystem model. We mostly focused on changes relative to the previously published Biome-
BGCMuSo v4.0 (Hidy et al., 2016), but our intention was also to provide a complete,
standalone reference for the modelling community with mathematical equations (detailed in
the Supplementary Material). Table 2 summarizes the structural changes that we made during
the developments starting from Biome-BGC v4.1.1 also including the previously published
Biome-BGCMuSo v4.0 (Hidy et al., 2016).

**Table 2. Comparison of model structural solutions for Biome-BGC 4.1.1, Biome-BGCMuSo v4.0 and Biome-**
**BGCMuSo v6.2.**

| Routine | original Biome-BGC | Biome-BGCMuSo v4.0 | Biome-BGCMuSo v6.2 |
|---|---|---|---|
| Runoff | no | based on simple, empirical formulation | distinguishing Hortonian and Dunne runoff |
| Pond water | no | simple solution | development of pond water formation (based on infiltration capacity) |
| Soil evaporation | Based on Penman-Monteith equation Calculation of the actual evaporation from the potential evaporation and the square root of time elapsed since the last precipitation. | Based on Penman-Monteith equation Calculation of the actual evaporation from the potential evaporation and the square root of time elapsed since the last precipitation. | Based on Penman-Monteith equation Parameterization possibility of actual aerodynamic resistance. Introduction of an upper limit for daily potential evaporation that is determined by the available energy. Calculation of the actual evaporation is based on the method Ritche (1981). Simulation of the reducing effect of surface residue or mulch cover on bare soil evaporation |
| Transpiration | Transpiration from one-layer bucket soil | Transpiration from 7-layers soil based on soil stress | Transpiration from 10-layers soil based on available water |
| Groundwater | no | Simple groundwater simulation. | Improvement of the simulation of groundwater effect (using capillary fringe). Introduction of two different methods. |
| Soil moisture stress | no | Relative SWC data is used to calculate soil water stress. The hygroscopic water, the wilting point, the field capacity and the saturation values of the soil layers can be defined in the input file layer by layer. The soil moisture stress index is affected by the length and the day since the drought event lasted. | The hygroscopic water, the wilting point, the field capacity and the saturation values of the soil layers can be defined in the input file layer by layer. The soil moisture stress index is affected by the length and the severity of the drought event, aggravated by the extreme temperature. Introduction of the soil curvature parameters to provide mechanism for soil texture dependent drought stress since it can affect the shape of the soil stress function. |



| | | | Normalized SWC data are used to calculate soil moisture stress index. |
|---|---|---|---|
| Organic carbon and nitrogen | One layer soil module with one organic carbon and nitrogen pool. | Multi-layered soil module without soil carbon and nitrogen profile. | Instead of defining a single litter, soil organic carbon and nitrogen pool, separate carbon and nitrogen pools for each soil layer in the form of soil organic matter and litter were implemented. Separation of above- and belowground litter pools. Litter and soil decomposition fluxes (carbon and nitrogen fluxes from litter to soil pools) are calculated layer by layer, depending on the actual temperature and SWC of the corresponding layers. Leaching of dissolved organic carbon and nitrogen. |
| Inorganic nitrogen | One layer soil module with one mineralized N pool. | Multi-layer soil module with an empirical inorganic N-profile (no layer-by-layer calculations, only estimation of the subpools in the different soil layer based on the rootlenght proportion). | Separation of ammonium (sNH4) and nitrate (sNO3) soil pools instead of a general mineralized nitrogen pool. Nitrification fluxes are calculated layer by layer, depending on the actual pH, temperature and SWC of the given layers. Denitrification fluxes are calculated layer by layer, depending on the depth, actual temperature and SWC of the given layers. |



Earlier model versions used a soil hydrology scheme based on the Richards equation, but the results were not satisfactory. Sándor et al. (2017) presented results from the first major grassland model intercomparison project (executed within the frame of FACCE MACSUR) where Biome-BGCMuSo 2.2 was used. That study demonstrated the problems associated with proper representation of soil water content that was a common shortcoming of all included models. In the Hidy et al. (2016) paper, where the focus was on Biome-BGCMuSo v4.0, the SWC related figures clearly indicated problems with the simulations compared to observations. The SWC amplitude was not captured well which clearly influences drought stress, decomposition, and other SWC driven processes like nitrification and denitrification. For the latter two processes this is especially critical as they are associated with contrasting SWC regimes (nitrification is an aerobic, while denitrification is an anaerobic process). This is a good example for erroneous internal process representation that may lead to improper results. Note that the currently used functions for nitrification/denitrification are also subject to uncertainty that needs to be addressed in the future (Heinen, 2006). Nevertheless, the





presented model developments might contribute to a more realistic soil process simulations
and improved results.
Algorithm ensemble approach is already implemented in Biome-BGCMuSo.
Algorithm ensemble means that the user has more than one option for the representation of
some processes. Biome-BGCMuSo v6.2 has alternative phenology routines (Hidy et al.,
2012), two alternative methods for soil temperature (Hidy et al., 2016), soil hydrology
(described in this study), photosynthesis and soil moisture stress calculation. We plan to
extend the algorithm ensemble by providing alternative decomposition schemes to the model.
One possibility is the implementation of a CENTURY-like structure (Koven et al., 2013) that
is a promising direction and might improve the quality of the equilibrium (spin-up)
simulations and the simulated N mineralization related to SOM decomposition. Reported
problems related to the rapid decomposition of litter in the current model structure (Bonan et
al., 2013) needs to be addressed in future model versions as well.
Plant growth and allocation related developments were not addressed in this study but
of course has many inferences with the presented model logic (i.e. parameterization and
related primary production defines the amount and quality of litter, etc.). A forthcoming
publication will provide a comprehensive overview on the plant growth and senescence
related model modifications where elements from crop models are also included.
Biome-BGCMuSo is still an open source model that can be freely downloaded from
its website with a detailed User's Guide and other supplementary files. We also encourage
users to test the so-called RBBGCMuso package (available at GitHub) that has many
advanced features to support model application and optimization. A graphical environment,
called AgroMo (also available at GitHub) was also developed around Biome-BGCMuSo to
help users in carrying out simulations either with site specific plot scale data or with gridded
databases representing large regions.

## Code and data availability

The current version of Biome-BGCMuSo, together with sample input files and detailed User's
Guide are available from the website of the model: http://nimbus.elte.hu/bbgc/download.html
under the GPL-2 licence. Biome-BGCMuSo v6 is also available at GitHub:
https://github.com/bpbond/Biome-BGC/tree/Biome-BGCMuSo_v6. The exact version of the
model (v6.2 alpha) used to produce the results used in this paper is archived on Zenodo



(https://doi.org/10.5281/zenodo.5761202). Experimental data and model parameterization used
in the study are available from the corresponding author upon request.


## Authors' Contributions

Hidy developed Biome-BGCMuSo, maintained the source code and executed the sample
simulations. The study was conceived and designed by Hidy, Barcza and Fodor, with
assistance from Ács, Dobor and Hollós. It was directed by Hidy and Barcza . Ács and Dobor
contributed with model benchmarking. Hollós participated with the construction of a
modeling framework for Biome-BGCMuSo.  Filep, Incze, Zacháry and Pásztor contributed
with experimental data. Hidy, Barcza, Fodor and Merganičová prepared the manuscript and
the supplement with contributions from all co-authors. All authors reviewed and approved the
present article and the supplement.


## Acknowledgements

The research was funded by the Széchenyi 2020 programme, the European Regional
Development Fund and the Hungarian Government (GINOP-2.3.2-15-2016-00028). Also
supported by grant "Advanced research supporting the forestry and wood-processing sector´s
adaptation    to    global    change    and    the    4th    industrial    revolution",    No.
CZ.02.1.01/0.0/0.0/16_019/0000803 financed by OP RDE". KM was also financed by the
project: "Scientific support of climate change adaptation in agriculture and mitigation of soil
degradation"  (ITMS2014+  313011W580)  supported  by  the  Integrated  Infrastructure
Operational Programme funded by the ERDF. We are grateful to Galina Churkina for
reviewing this manuscript.

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
