# Peer review of "Soil related developments of the Biome-BGCMuSo v6.2"

_Geoscientific Model Development, 2021_

## Community Comment (CC4)

[revised manuscript text omitted]

**10. Land use map of Hungary**

Land use maps were created based on the CORINE-2018 database (EEA, 2022) using 25 ha minimum mapping units. Its 10 km spatial resolution version, that is in conform with the soil and weather databases used in the study, was created by selecting the dominant land use type for each cell after counting the number of mapping units within the cell in each land use category.

Fig. S1 shows the spatial distribution of croplands, grasslands and forests for Hungary in 100m resolution based on the CORINE-2018 database (EEA, 2022) using 25 ha minimum mapping units.

[Figure]

**Figure S1: Fine resolution (25 hectare mapping units) land use map of Hungary based on the CORINE-2018 database.**

Fig. S2 shows the 10 km resolution version of the land use map that is in conform with the soil and weather databases used in the study. The map was created by selecting the dominant land use type for each cell after counting the number of mapping units within the cell in each land use category.

[Figure]

**Figure S2: The spatial distribution of the dominant land use types using the grid resolution (approximately 10 km) that is defined for the country-wide simulations in the study (see main text).**

**11. Observation based soil organic carbon content of the topsoil in Hungary**

[Figure]

**Figure S3: Fine resolution (100 m) map of topsoil (0-30 cm depth) soil organic carbon (SOC) content in Hungary based on the DOSoReMI database (Pásztor et al., 2020).**

Fig. S3 shows the observation based SOC map of Hungary based on the DOSoReMI database (Pásztor et al., 2020). In order to harmonize the soil input data grid with weather data grid the 100 m resolution soil data were upscaled. For every approximately 10×10 km grid cells one representative cell out of the 10.000 constituting 100×100 m subcells of the DOSoReMI grid (Pásztor et al., 2020) has been selected using a dominant soil type method proposed by Fodor et al. (2021). The soil organic carbon contents of the topsoil of the representative cells are shown in Figure S4.

[Figure]

**Figure S4: 10 km resolution map of topsoil (0-30 cm depth) soil organic carbon (SOC) content in Hungary.**

---

## Author Response (AR1)

**Reply to the interactive comment**
**RC1: 'Comment on gmd-2021-389', Anonymous Referee #1, 05 Jan 2022**

First of all, we thank Anonymous Referee #1 for reviewing the manuscript (gmd-2021-389). Thank you very much for the positive words. Here we answer the questions and issues raised by the Reviewer in detail. We also attached the manuscript (MS) and the supplementary material revised in the light of the comments received.

Note that the comments of Anonymous Referee #1 are shown below *in italic*. Our responses to the comments are presented below in normal font style.

*In **Soil related developments of the Biome-BGCMuSo v6.2 terrestrial ecosystem model by integrating crop model components** the authors presents the latest implementations into the Biome-BGC model. The study is a good example of how to do a model description but maybe less so for model evaluation. The text and description (equations and simulation setup) are easy to read and follow.*

Indeed, our intention was to present a detailed overview about the latest developments. Case studies are presented to highlight some features that might deserve attention by the community. It was not our intention to present detailed model evaluation here, partly because of length restrictions, partly because it was out of scope of the MS. Nevertheless, based on the suggestion and also according to the suggestion of Reviewer #2 we extended the MS with one additional case study focusing on soil N cycle and soil respiration. Forthcoming studies will provide further model evaluation.

*I have two things that I think should be addressed: first the title, it mentions crops but the crop specific components are not that visible in the manuscript. This could be addressed by showing for instance the geography of cropland land management types in the data used. At the moment it's really hard to get from the results how the model are performing on different land covers. And maybe also showing the representative land cover classes on a map.*

It is possible that this could be a misunderstanding. Croplands and other land use types are not specifically addressed in the study. We did not mention cropland specific investigation in the original manuscript with the exception of the case study of the lysimeter station (but this is a local scale simulation). The title referred to crop models since we took important features from crop models and implemented them in the Biome-BGCMuSo which is/was a general purpose biogeochemical model missing these options before. With the improvements, Biome-BGCMuSo can simulate almost all kinds of terrestrial ecosystems (forests, shrubs, grasslands and croplands). In the manuscript, we demonstrated that it is really useful to import features from crop models and use them in biogeochemical models. In the manuscript we mention the crop model elements explicitly (e.g. lines 271-273 and 334-337 in the revised MS).

In any case, as a response to the other issue raised by the Reviewer we present a land use map and a soil organic carbon (SOC) map in the end of the revised Supplementary Material.

*This brings me on to the second part, although it is interesting to see how the updated parameters increased the overall model performance, but what are the simulated soil carbon stocks? I guess the evaluation of other components such as yields and aboveground biomass as well as fluxes will be presented in the following publication.*

We have included the observation based SOC map in the revised Supplementary Material (see previous answer). The SOC map is derived from the DOSoReMI database (Pásztor et al., 2020). See reference in the Reference section of the MS. We started to work on a manuscript that will present the plant related developments of the model (this one focuses on the soil related developments). Validation of the model for eddy covariance based carbon fluxes and yield will be presented in that MS. Evaluation of soil $N_2O$ and $CO_2$ efflux simulations were included in the revised manuscript as a third case study based on observations.

*Some specific comments on the text:*

*Line 105: the supplied reference for this statement is pretty old, LPJ-mL, LPJ-GUESS, Orchidee, etc. are all models that have implemented this after that publication.*

We made a mistake during the MS editing, as originally we referred to a recent publication (Friedlingstein et al., 2020) but this was somehow modified to Friedlingstein et al. (2010) (which was in fact not in the References) which has no Table A1 at all. The Friedlingstein et al. (2020) paper contains the mentioned Table A1 which supports the statement. It is clear from the table that some DGVMs still miss some management options. Although the 2020 paper is quite recent we made a quick literature search to see the present situation. It is obvious that some models listed in Table A1 are still missing some management options (while e.g. ORCHIDEE was extended with grazing and irrigation options; grazing was added to LPJ-GUESS; CLM now has a fertilization module). Anyway, we have slightly modified our statement maintaining our original position on the matter. See lines 108-111 in the revised MS that reads the following:

"Additionally, human intervention representation (management) is still incomplete in some state-of-the-art BGMs, e.g. thinning, grass mowing, grazing, tillage or irrigation is missing in some models (see Table A1 in Friedlingstein et al., 2020)."

*Line 175-178: This is not novel for Biome-BGC, this sounds very much like what is carried out in LPJ-GUESS.*

What we meant here is that this is novel in the Biome-BGC 'universe'. We extended the sentence to avoid this misunderstanding. In the revised MS (lines 178-181 in the revised MS) it reads:

"So-called transient simulation option (which is the extension of the spinup routine) is a novel feature in Biome-BGCMuSo v6.2 relative to the previous versions in order to ensure smooth transition between the spinup and normal phase (Hidy et al., 2021)."

*Line 714: Wrong model version?*

Corrected. Indeed, it should be 4.0.

*Line 747-755: This whole paragraph is unfinished, especially which part is concerning grasslands and forests.*

Thank you for finding this error. Indeed, the text was not entirely correct. Now we reformulated the paragraph (lines 839-848 in the revised MS):

"Grassland ecophysiological parameterization without management was used in the spinup phase to initialize SOC pools for croplands. For the transient phase cropland parameterization was used with fertilization, ploughing, planting and harvest settings. In case of grasslands, both during the spinup and transient phases grassland parameterization was used, and in the

transient phase mowing was assumed once a year. In case of forests generic deciduous broadleaf forest parameterization was used for both spinup and transient phases with thinning in the latter phase. For our parameterization presented in the MS the generic, plant functional type specific ecophysiological parameter sets published by White et al. (2000) served as starting points. These Biome-BGCMuSo specific parameter sets are available at the website of the model[1]."

**Reply to the interactive comment**
**RC2: 'Comment on gmd-2021-389', Anonymous Referee #2, 26 Jan 2022**

First of all, we thank Anonymous Referee #2 for reviewing the manuscript (gmd-2021-389). Thank you very much for the positive words. Here we answer the questions and issues raised by the Reviewer in detail.

We also attached the manuscript (MS) and the supplementary material revised in the light of the comments received.

Note that the comments of Anonymous Referee #2 are shown below *in italic*. Our responses to the comments are presented below in normal font style.

*This manuscript presents a detailed description of several recent improvements in the Biome-BGCMuSo v6.2 model, including soil hydrology-related processes and carbon/nitrogen-related processes. Overall, this manuscript is easy to follow and these model improvements are worth publishing, which may provide reference for the development of relevant terrestrial biosphere models. However, some parts of the manuscript read more like a technical report than a research paper, and I would like to suggest modifying relevant parts (I point out some). Meanwhile, the authors may want to add other experiments to show the overall performance of the new model in simulating nitrogen-related variables (e.g., N2O or N leaching). Please find my specific comments below.*

After reading numerous similar papers, we tend to think that almost all papers focusing mainly on model development seem to be a bit too technical, more like an inventory of the improvements. This is probably due to the complexity of those models. Nevertheless, we think that development papers are essential to provide theoretical basis promoting the wider application of the model.

Indeed, case studies are more attractive for the general audience. According to the comments of the Reviewer we extended the manuscript with an additional case study addressing the simulated soil nitrogen cycle including soil $N_2O$ efflux, supplemented by soil respiration simulations. As the present study focuses on soil processes, vegetation related carbon fluxes (gross primary production, net ecosystem exchange, etc.) are not addressed. We are working on a second manuscript that will present the plant related developments of Biome-BGCMuSo where ecosystem scale carbon flux simulations will be validated by using eddy-covariance data. We do hope that the Reviewer will accept our decision and will find the modifications sufficient.

*Line 1-3: You may need to change the title of the manuscript, as the current one puts too much emphasis on crop model and may be confusing.*

We realized that the original title can be misleading, so we changed it by leaving out mentioning the crop model connection. The new title is "Soil related developments of the Biome-BGCMuSo v6.2 terrestrial ecosystem model"

*Line 37-39: You may need to revise this sentence since it reads more like a technical report than a research paper*

We simply deleted the sentence. It really does not add much to the message.

*Line 117: Maybe delete this sentence*

We deleted the sentence.

*Line 123: What management practices?*

We have extended the sentence (lines 126-130 in the revised MS):
"Biome-BGCMuSo v4.0 (Biome-BGC with Multilayer Soil module) uses a 7-layer soil module and is capable of simulating different ecosystems from natural grassland to cropland including several management options (mowing, grazing, thinning, planting and harvest), taking into account several environmental effects (Hidy et al., 2016)."

*Line 235, Line 332, Line 337......: You may need to change such expressions: "An important novelty ......"*

Thank you for the remarks. We have modified the text accordingly.
We have changed the text (line 235 in the original MS; lines 238-239 in the revised MS): "A new development in Biome-BGCMuSo v6.2 is that maximum infiltration is calculated based on the saturated hydraulic conductivity and the SWC of the top soil layers."

We have also replaced the expression "*important novelty*" (Line 332 in the original MS; lines 334-336 in the revised MS): "A new feature in Biome-BGCMuSo v6.2 is the calculation of the actual evaporation from the potential evaporation and the square root of time elapsed since the last precipitation (expressed by days; Ritchie, 1998)."

We have modified the sentence in line 337 of the original MS (lines 339-340 in the revised MS): "One major new development in Biome-BGCMuSo v6.2 is the simulation of the reduction effect of surface residue or mulch cover on bare soil evaporation."

*Line 345-347: Did you consider the thickness of the residues in your model?*

Not explicitly. We calculate surface coverage that could be larger than 100%. The increasing surface cover values above 100% are analogous and have a similar effect to the increasing thickness of residues.

*Line 369-370: How did you consider/calculate the distribution of root density?*

Biome-BGCMuSo v6.2 calculates the vertical distribution of roots according to Jarvis' method (1989). The relative amount of roots ($R_i$) in a layer is estimated with the following equation:

$$R_i = f\left(\frac{\Delta z_i}{z_r}\right) e^{-f\left(\frac{z_i}{z_r}\right)}$$

Where $f$ is an empirical root distribution parameter, $\Delta z_i$ and $z_i$ are the thickness and mid-point depth below the soil surface of layer $i$ respectively, and $z_r$ is depth expressed in dimensionless terms as a fraction of the root depth. This development was already published in Hidy et al. (2016). We included the citation to the Jarvis paper in the MS.

*Line 398: In this section, the effects of water scarcity stress on plant physiological processes have been considered, but, how did you consider the water logging effects? Excessive water in the soil could also significantly affect various plant/soil-related processes.*

In Biome-BGCMuSo water logging negatively affects transpiration and assimilation, but increases senescence and denitrification. A critical SWC parameter was introduced to take this phenomenon into account: if actual SWC is greater than this critical value, a soil stress factor is calculated using a linear ramp function that is zero at this critical value (limitation starts) and equals to 1 at saturation (complete limitation). This function is presented in the manuscript (Eq. 5). We also addressed this topic in our previous publication (Hidy et al., 2016).

*Line 481: Typically, plant nitrogen fixation process is controlled by many environmental factors, including soil moisture, substrate concentration, soil temperature, etc. You may want to incorporate the relevant N-fixation processes in the model to expand its application capabilities on the regional scale.*

We agree with the Reviewer, N fixation is a relevant topic. In the current version of Biome-BGCMuSo a single, user adjustable parameter is provided in the ecophysiological input file to define the fixation rate in $kgN/m^2/y$ uniformly distributed throughout the year. So, at the moment, no environmental effect is taken into account. We are in the process of taking stocks of the options proposed in the literature. In the future we plan to incorporate the FUN (sub)model of Brzostek et al. (2014) that would be a possible solution for a more sophisticated estimation of plant-mycorrhizal fungi interactions. Another proposed method can be found in Thomas et al. (2013) that is readily available for the user for a rough estimation based on simple climate data.

*Line 527: I think you may want to use "then" instead of "than".*

Yes, the sentence has been corrected.

*Line 528: Are the C:N ratios the same for different soil carbon pools? Dynamic or static?*

The simple answer is that they are static. The C:N ratio of the passive SOC pool (S4 in the model) can be set by the user in the soil input file and the C:N ratio of the other SOC pools are calculated from this input data following the logic defined in the original Biome-BGC model. We mention the nature of the C:N ratios is the MS (lines 524-533 in the revised MS) that reads the following:
"In the original Biome-BGC and in previous Biome-BGCMuSo versions the C:N ratio (CN) of the soil pools were fixed in the model code. In Biome BGCMuSo v6.2 the CN of the passive soil pool (S4 in Figure 4; passive soil organic matter) can be set by the user as a soil input parameter. The CN of the other soil pools (labile, medium and slow; S1, S2 and S3) are calculated according to the following proportions defined in the original Biome-BGC model: $CN_{labile} / CN_{passive} = 1.2$; $CN_{medium} / CN_{passive} = 1.2$; $CN_{slow} / CN_{passive} = 1$. The CN of the donor and acceptor pools are used in decomposition calculations (see details in Supplement Material, Section 7). The donor and acceptor pools can be seen in Figure 3 and Figure 4."

*Line 726: In addition to SOC, I would like to see the simulation performance of CO2 flux and compare these simulations with flux-based observations.*

As we mentioned above, vegetation related carbon fluxes (gross primary production, net ecosystem exchange, etc.) are not addressed in this study. A forthcoming manuscript will present the plant related developments of the model where ecosystem scale $CO_2$ flux simulations will be validated. We refer here to Fodor et al. (2021) where validation of the model is presented for the Klingenberg (DE-Kli1) eddy covariance site (see Supplementary

material of Fodor et al., 2021; https://doi.org/10.1080/17538947.2021.1953161). This reference has been added to the MS as well (lines 763-764 in the revised MS). Soil $CO_2$ efflux simulations are also compared to observations in the revised MS (lines 795-806 in the revised MS).

*Line 647: Since nitrogen cycle is a major improvement in your model, I suggest adding other experiments to show the simulation performance of nitrogen-related variables (e.g., N2O, N loading, etc.).*

We added a new case study presenting N cycle related simulation results (5.2 Evaluation of the soil nitrogen balance module and the simulated soil respiration; lines 732-815 in the revised MS).

**References**

Brzostek, E. R., Fisher, J. B., Phillips, R. P.: Modeling the carbon cost of plant nitrogen acquisition: Mycorrhizal trade-offs and multipath resistance uptake improve predictions of retranslocation, Journal of Geophysical Research: Biogeosciences, 119, 1684–1697, https://doi.org/10.1002/2014JG002660, 2014.

Fodor, N., Pásztor, L., Szabó, B., Laborczi, A., Pokovai, K., Hidy, D., Hollós, R., Kristóf, E., Kis, A., Dobor, L., Kern, A., Grünwald, T., Barcza, Z.: Input database related uncertainty of Biome-BGCMuSo agro-environmental model outputs, International Journal of Digital Earth, 14, 1582–1601, https://doi.org/10.1080/17538947.2021.1953161, 2021.

Thomas, Q., Bonan, G., Goodale, C.: Insights into mechanisms governing forest carbon response to nitrogen deposition: A model–data comparison using observed responses to nitrogen addition, Biogeosciences 10, 3869–3887, https://doi.org/10.5194/bg-10-3869-2013, 2013.